# Oxidation processes in the Eastern Mediterranean atmosphere: Evidence from the Modelling of HO$_x$ Measurements over Cyprus

C. Mallik[1], L. Tomsche[1], E. Bourtsoukidis[1], J.N. Crowley[1], B. Derstroff[1], H. Fischer[1], S. Hafermann[1], I. Hueser[1], U. Javed[1], S. Kessel[1], J. Lelieveld[1,2], M. Martinez[1], H. Meusel[3], A. Novelli[4], G. J. Phillips[5], A. Pozzer[1], A. Reiffs[1], R. Sander[1], D. Taraborrelli[4], C. Sauvage[1], J. Schuladen[1], H. Su[2], J. Williams[1,2] and H. Harder[1].

[1]Atmospheric Chemistry Dept., Max Planck Institute for Chemistry, Mainz, 55128, Germany

[2]Energy, Environment and Water Research Center, Cyprus Institute, Nicosia, 1645, Cyprus

[3]Multiphase Chemistry Dept., Max Planck Institute for Chemistry, Mainz, 55128, Germany

[4]Institute of Energy and Climate Research, Forschungszentrum Juelich GmbH, Julich, 52425, Germany

[5]Department of Natural Sciences, University of Chester, Chester, CH1 4BJ, United Kingdom

*Correspondence to:* Hartwig Harder (hartwig.harder@mpic.de) & Chinmay Mallik (chinmay.mallik@mpic.de)

**Abstract**

The Mediterranean is a climatically sensitive region located at the crossroads of air masses from three continents: Europe, Africa and Asia. The chemical processing of air masses over this region has implications not only for the air quality, but also for the long-range transport of air pollution. To obtain a comprehensive understanding of oxidation processes over the Mediterranean, atmospheric concentrations of the hydroxyl radical (OH) and the hydroperoxyl radical ($HO_2$) were measured during an intensive field campaign (CYprus PHotochemistry EXperiment, CYPHEX-2014) in the north-west of Cyprus in the summer of 2014. Very low local anthropogenic and biogenic emissions around the measurement location provided a vantage point to study the contrasts in atmospheric oxidation pathways under highly processed marine air masses and those influenced by relatively fresh emissions from mainland Europe.

The CYPHEX measurements were used to evaluate OH and $HO_2$ simulations using a photochemical box model (CAABA/MECCA) constrained with CYPHEX observations of $O_3$, CO, $NO_x$, hydrocarbons, peroxides and other major $HO_x$ (OH + $HO_2$) sources and sinks in a low $NO_x$ environment (<100 pptv NO). The model simulations for OH agreed to within 10 % with *in situ* OH observations. Model simulations for $HO_2$ agreed to within 17 % of the *in situ* observations. However, the model strongly underpredicted $HO_2$ at high terpene concentrations, this underprediction reaching up to 38 % at the highest terpene levels. Different schemes to improve the agreement between observed and modelled $HO_2$, including changing the rate coefficients for the reactions of terpene generated peroxy radicals ($RO_2$) with NO and $HO_2$ as well as the autoxidation of terpene generated $RO_2$ species, are explored in this work. The main source of OH in Cyprus was its primary production from $O_3$ photolysis during the day and HONO photolysis during early morning. Recycling contributed about one-third of the total OH production, and the maximum recycling efficiency was about 0.7. CO, which was the largest OH sink was also the largest $HO_2$ source. Lowest $HO_x$ production and losses occurred when the air masses had higher residence time over the oceans.

Keywords: hydroxyl radical (OH), hydroperoxyl radical ($HO_2$), terpene chemistry, peroxy radical ($RO_2$), box model (CAABA/MECCA), oxidation capacity, chemical cycling, OH-$HO_2$ budgets

## 1. Introduction

1.1. Air pollution and $HO_x$ chemistry

The chemical and photochemical processing of air pollutants, in conjunction with local emissions, meteorology and atmospheric transport, strongly influences the air quality over a region. The regional air quality impacts human health, agriculture, the overall condition of the biosphere and subsequently the climate. Studies attribute 2-4 million premature deaths globally to outdoor air pollution (Silva et al., 2013; Lelieveld et al., 2015). Oxidants in the Earth's atmosphere prevent the pollutants released into it from building up to toxic levels. These oxidants not only convert many toxic pollutants into less toxic forms e.g. carbon monoxide (CO) to carbon dioxide ($CO_2$) but also help in their removal e.g. nitrogen oxides ($NO_x$) and sulfur dioxide ($SO_2$) are converted into soluble nitric acid ($HNO_3$) and sulfuric acid ($H_2SO_4$) respectively, although some toxic chemicals may still be formed during degradation of non-toxic ones. Most of the oxidation processes in the atmosphere proceed along reaction pathways initiated by the hydroxyl radicals (OH) during the day (Levy, 1971), making it the dominant chemical cleaning agent of the daytime atmosphere (Lelieveld et al., 2004). The dominant primary production of OH is via the photolysis of ozone ($O_3$) at ultraviolet wavelengths (Crosley, 1995) producing the electronically excited $O(^1D)$ atoms (R1). $O^1D$ that escapes quenching (R2) reacts with water vapor in the atmosphere to produce OH radicals (R3)

$$O_3 + h\nu \rightarrow O(^1D) + O_2 \qquad\qquad\qquad (R1, \lambda<330 \text{ nm})$$

$$O(^1D) + M \rightarrow O(^3P) + M \ (M = N_2, O_2) \qquad\qquad (R2)$$

$$O(^1D) + H_2O \rightarrow 2OH \qquad\qquad\qquad (R3)$$

Because OH reacts rapidly with several biogenically and anthropogenically emitted gases (e.g. R4), its lifetime is typically a few hundred milliseconds in the troposphere. Oxidation limits these gases (methane ($CH_4$), CO, volatile organic compounds) from accumulating above trace levels in the troposphere. Reactions of OH with CO and hydrocarbons (R4) also trigger quasi-instant formation of the hydroperoxyl radical ($HO_2$, via R5) in our $O_2$ rich atmosphere. In the presence of nitric oxide (NO) and $O_3$ (R6, R7), a chemical equilibrium is set between OH and $HO_2$ on a timescale of seconds. The sum of OH and $HO_2$ is known as $HO_x$. The reactions R6 and R7 maintain the oxidizing efficiency of the atmosphere by recycling OH. Simultaneously, in the presence of $NO_x$ (=NO + nitrogen dioxide ($NO_2$)), they also propel a reaction chain leading to $O_3$ formation via R8 and R9. R8-R9 are the main chemical source of tropospheric $O_3$, which is an important oxidant, a greenhouse gas as well as a major secondary pollutant.

$$CO + OH \rightarrow CO_2 + H \qquad\qquad\qquad (R4)$$

$$H + O_2 \ (+M) \rightarrow HO_2 \ (+M) \qquad\qquad\qquad (R5)$$

$$NO + HO_2 \rightarrow NO_2 + OH \qquad\qquad\qquad (R6)$$

$$HO_2 + O_3 \rightarrow OH + 2O_2 \qquad\qquad\qquad (R7)$$

$$NO_2 + h\nu \rightarrow O(^3P) + NO \qquad\qquad\qquad (R8, \lambda<430 \text{ nm })$$

$$O(^3P) + O_2 (+M) \rightarrow O_3 (+M) \qquad\qquad (R9)$$

OH also oxidizes anthropogenic and biogenic hydrocarbons (HCs) emitted into the atmosphere leading to peroxy radicals ($RO_2$). NO converts the $RO_2$ into $HO_2$, which again leads to OH recycling (R6). The $HO_x$ cycle can be terminated by the radical+radical interactions of $RO_2$ and $HO_2$ with and between themselves to form peroxides (R10, R11). Depending on NO levels, the $RO_2$ species can react through different channels. Under low $NO_x$ conditions e.g. in clean marine boundary layer, when the reaction of $RO_2$ with NO is less significant than its reaction with $HO_2$, peroxide formation becomes a major sink for $HO_2$, further leading to OH and $O_3$ removal. Under higher NO levels, $RO_2$ would react with NO to form $NO_2$ ($O_3$ source), $HO_2$ (OH source), alkyl nitrates as well as aldehydes (R12, R13, R14). Further, there have been suggestions of auto-oxidation of $RO_2$ to produce peroxides and yielding OH under atmospheric conditions (Crounse et al., 2013). These auto-oxidation mechanisms proceed through catalytic H-shifts at a very fast rate (up to ~ 0.1 $s^{-1}$) e.g., conversion of $RO_2$ generated from a carbonyl compound into a highly functionalized dicarbonyl hydroperoxide compound (R15).

$$HO_2 + HO_2 \rightarrow H_2O_2 + O_2 \qquad\qquad (R10)$$

$$RO_2 + HO_2 \rightarrow ROOH + O_2 \qquad\qquad (R11)$$

$$RO_2 + NO + O_2 \rightarrow R'CHO + HO_2 + NO_2 \qquad\qquad (R12)$$

$$RO_2 + NO \rightarrow RNO_3 \qquad\qquad (R13)$$

$$NO_2 + OH + M \rightarrow HNO_3 + M \ (M = N_2, O_2) \qquad\qquad (R14)$$

In polluted air when $NO_x$ concentrations become large enough that the reaction of $NO_2$ with OH dominates over other $HO_x$ sinks, the major loss of $HO_x$ is via the formation of nitric acid ($HNO_3$, R14), which is then removed from the atmosphere by wet and dry deposition. Some acid forming reactions are also the starting point of particle formation in the atmosphere e.g. via oxidation of $SO_2$. In sum, oxidation is a means by which major pollutants are processed and removed from the atmosphere, but with concomitant implications to air quality, agriculture, climate and health. However, large uncertainties in atmospheric reaction pathways, rate coefficients and measurements of atmospheric gases and radicals, limit our ability to understand and predict the quality of ambient air.

1.2. Motivation for the campaign

The Mediterranean is a climatically sensitive region which is rapidly getting warmer and dryer over time (Lelieveld et al., 2012). The atmospheric chemistry over the region is complex with regular exceedances of the European Union (EU) ozone air quality standard (Doche et al., 2014; Kalabokas et al., 2013) and a projected increase of summertime

$O_3$ over the Eastern Mediterranean (Lelieveld et al., 2002; Lelieveld and Dentener, 2000). Cyprus is a small, remote island (area: 9251 $km^2$; population: 1.15 x $10^6$; Density: 119.2 people/$km^2$) in the Eastern Mediterranean (south of mainland Europe) with very low local emissions. The non-methane volatile organic compounds (NMVOCs), $NO_x$, $SO_x$ emitted by Cyprus and European Union (EU28) were 6.8, 17.2, 16.8 Gg and 6722, 7819, 3083 Gg respectively for 2014 (European Union emission inventory report, 2017). Being located downwind of the mainland Europe emissions, Cyprus provides an ideal vantage point to study the impact of European emissions on the Mediterranean climate (Lelieveld et al., 2002). To study the atmospheric chemistry over the Mediterranean and understand the impact of emissions from different regions of Europe on air quality in Cyprus, an intensive field measurement campaign (the Cyprus Photochemistry EXperiment, hereafter CYPHEX) was conducted in Cyprus during the summer of 2014, wherein a comprehensive suite of trace gases was measured.

Cyprus is exposed to air masses influenced by emissions from South-West European countries which are photochemically processed over the Mediterranean Sea. The Etesian winds during summer carry air masses loaded with emissions from industry and biomass burning over Central/East European countries.  Although emissions from Europe have been steadily decreasing over the past few decades (e.g. $NO_x$, $SO_x$ and NMVOCs decreased by nearly 50 %, 80 % and 57 % respectively during 1990-2013; European Union emission inventory report, 2017), these emissions are still substantial w.r.t. the global total. Apart from the influence of European emissions, Cyprus is strategically located at cross-roads of air masses from Asia, Africa and the Atlantic (Kleanthous et al., 2014). Influence from all these different source regions, incorporating the impact of a plethora of hydrocarbons, nitrogen and sulfur species as well as mineral dust on the  atmospheric processing in this part of the world,  have attracted several field campaigns (Carslaw et al., 2001; Berresheim et al., 2003). For example, the Mediterranean INtensive Oxidant Study (MINOS) was conducted to study the budget of atmospheric oxidants influenced by long-range transport of pollution, in the summer of 2001 (Lelieveld et al., 2002; Gros et al., 2003; Salisbury et al., 2003). OH radicals were measured in the coastal boundary layer of Crete as part of the MINOS campaign and the major sources of OH were found to be photolysis of $O_3$ and recycling from $HO_2$+NO reaction (Berresheim et al., 2003). Although $HO_2$ was not measured during MINOS, high levels of formaldehyde (HCHO), a major primary $HO_2$ source, were observed (Kormann et al., 2003). In the present study, we report simultaneous measurements of OH and $HO_2$ during the CYPHEX campaign, and compare these observations with simulations from a photochemical box model constrained with $O_3$, CO, NO, $NO_2$, peroxides, several anthropogenic and biogenic hydrocarbons including important alkanes, alkenes, alkynes, aldehydes and acids and photolysis frequencies of important gases relevant to $HO_x$ chemistry. The main objective of this study is to identify the major chemical species and reaction pathways controlling the $HO_x$ chemistry over this climatically sensitive region.

## 2.  Methods

2.1 Measurement site

The CYPHEX campaign was conducted at the northwest coast of Cyprus, on a hilltop about 650 m AMSL, in Paphos District of the island and facing the sea about 5-8 km away on the west. The Akamas peninsula national park is about 25 km to the northwest of the measurement site (34° 57' N / 32° 23' E), with terrain descending rapidly to the northeast

towards the town of Polis, with a population of about 2000. The area comprises of spare shrub type vegetation with small Mediterranean trees comprising pines, junipers, olives, carob, pomegranates, and almonds. The area is weakly populated in a radius of about 20 km. The site is away from the major cities: Paphos with 90000 people is 20 km south, Limassol with 235000 people is 70 km south east, Larnaca with 145000 people is 110 km east and Nicosia with

325000 people is 90 km northeast (census of population Oct 2011; www.cystat.gov.cy). The local wind direction is predominantly southwest (70 % between $202.5°$ -$247.5°$) (Meusel et al., 2016; Derstroff et al., 2017), bringing in humid air from the sea, with no immediate anthropogenic influence. Analysis of 5 day back trajectories show mainly two major air mass regimes (Huser et al., 2017). Etesian winds influenced by fresh emissions from East-Central Europe and crossing Turkey and Greece, and Mistral winds influenced by emissions from South-West Europe but

processed for a longer period over the Mediterranean (Figure 1). During the summer of 2014, a weakened eastwest pressure gradient led to weak and delayed Etesian winds (Tyrlis et al., 2015). The site is also influenced by local land sea breezes.

2.2 $HO_x$ measurements during CYPHEX

Atmospheric OH and $HO_2$ were measured during CYPHEX using the HydrOxyl Radical measurement Unit based on

fluorescence Spectroscopy (HORUS) instrument (Martinez et al., 2010). The setup was based on the well-established Laser Induced Fluorescence-Fluorescence Assay by Gas Expansion (LIF-FAGE) technique for atmospheric OH measurements (Brune et al., 1995; Crosley, 1995; Hard et al., 1984). The laser system consisted of a tunable dye laser which was pumped by a diode-pumped Nd:YAG laser (Navigator I J40-X30SC-532Q, Spectra Physics) pulsing at 3 kHz. The two laser assemblies were mounted on either side of a vertical plate, which was mounted on a rack inside a

measurement container about 6 m x 2.5 m x 2.5 m. The output laser radiation was split in a 9:1 ratio using beam splitters, and channeled through 8 m optical fibers into the detection module for measurements of OH and $HO_2$.

The detection system was mounted about 6 m above ground level (AGL) on the top of a scaffolding tower, constructed close to the wall of an air-conditioned container adjacent to the Laser unit. The container with the HORUS instrument was mounted on top of another similar measurement container with $O_3$, CO, $NO_x$ and peroxide instruments inside.

The inlets for all the instruments measuring during CYPHEX were placed within 5 m of the HORUS detection unit on the tower. Ambient air was drawn in HORUS through a critical orifice with pinhole size of 0.9 mm into a detection cell with about 4.3 mbar pressure. This internal pressure was achieved by using a combination of a roots blower (M90 compressor from Eaton, USA) followed by a scroll pump (XDS 35i, Edwards, USA). The resulting high volume flow ensured that the exchange of air illuminated by the laser is fast enough between two consecutive pulses, thus avoiding

possible laser-induced interferences.

The interaction of the ambient air with the laser beam in the low pressure detection cell was maximized using a White cell setup, where the laser light was reflected 32 times across the detection volume. The OH molecules were selectively excited by 308 nm laser light pulsing at 3 kHz on resonance with the $Q_1(2)$ transition line ($A^2\Sigma–X^2\prod$, v'=0 ←v''=0). An etalon controlled by a stepper motor, in the dye laser setup, caused the laser radiation to be tuned on and off

resonance with the OH transition every 7 s to account for the OH fluorescence plus background signals and the

background signals, respectively, resulting in a time resolution of 14 s. The spectra from a reference cell, where OH was produced by $H_2O$ thermolysis, was used to detect the $Q_1(2)$ line. The counts from the reference cell were used to sustain the on-resonance position while the etalon tethered around it. Although the pressure reduction in the detection cell results in smaller numbers of OH molecules, it improves the OH fluorescence quantum yield due to reduced collisional quenching of excited OH radicals. The lifetime of the excited OH extends beyond the Mie scattering by aerosols and the Raleigh scattering by smaller molecules/particles. These interfering background signals were discriminated by using time-gated micro-channel plate detectors (Hamamatsu), enabling excitation of OH radicals and detection of fluorescence at the same wavelength. The excitation at 308 nm is advantageous to the excitation at 282 nm as interference from laser generated OH from $O_3$ photolysis is ~25 times smaller at 308 nm. A band pass filter with a central wavelength of $308 \pm 0.75$ nm and bandwidth of 5 nm (FWHM) in front of the detector prevented signals due to any interfering wavelengths from registering on the detector. The fluorescence decay was recorded with a time-resolution of 4 ns and integrated for on-resonance and off-resonance periods, thus delineating the signal from the spectroscopic background. Further, the spectra of the measured atmospheric OH was compared with the one obtained from a reference cell ensuring no interferences from fluorescence of species like $SO_2$ and naphthalene.

The chemical background, originating from OH being internally produced during the transit time of about 3 ms from the orifice to the center of the detection cell, was accounted for by removing ambient OH with an OH scavenger. This scavenger, 3-5 sccm of pure propane in a carrier flow of 8000 sccm synthetic air for CYPHEX, was introduced through an inlet pre-injector (IPI) mounted on top of the inlet nozzle (Novelli et al., 2014a; Hens et al., 2014). The scavenger amount was just sufficient to scavenge off ~95% of atmospheric OH as determined from propane titration experiments on-site. IPI was connected to a blower that draws about 180-200 lpm of ambient air from the top of the IPI, the flow being monitored by a differential pressure sensor. The OH scavenger was injected via eight 0.5 mm diameter holes 5 cm above the pinhole of the OH inlet into the center of the air flow sampled by IPI close to its hyperbolic minima (Novelli et al., 2014b). The hyperbolic internal shape of the IPI (max cross section of 35 mm and min cross section of 6 mm) helps to sample air that has minimum wall contact and flows at a high velocity through a small diameter ensuring that the scavenger is well mixed with atmospheric air. The high flow also ensures minimum impact of the horizontal wind speed on the mixing efficiency. The IPI cycling was automated by a script resulting in repeated cycles of scavenger injection, flushing and no injection. The scavenger was periodically injected every 2 min, resulting in alternating measurements of background OH and total OH i.e. with and without scavenger injection. The difference between these two signals gives a measure of the atmospheric OH with a time resolution of 4 min. The wall losses in the IPI were periodically determined by physically dismounting the IPI for 5 min during measurements during different times of the day. On average, the background signal was 45 % of the total signal during daytime (Supplementary Figure 1). While the constituents of background OH can have pronounced impact on atmospheric oxidation processes, especially over regions with high biogenic emissions (Mauldin et al., 2012), the study of the influence of background OH on oxidation pathways during CYPHEX is beyond the scope of the present study.

Atmospheric $HO_2$ was measured in a detection cell located 16 cm below the OH detection block. $HO_2$ measurements were achieved by injecting NO purified through a sodium hydroxide coated silica (Sigma-Aldrich Ascarite) using a

1/8'' od, 0.035'' id coiled stainless steel loop with about 3 cm diameter, placed just above the $HO_2$ detection block. The NO was injected through 0.1 mm holes, leading to conversion of $HO_2$ into OH and detected by the same principle as in the first block. To minimize impact of $RO_2$ on OH formation (Fuchs et al., 2011; Hens et al., 2014), only about 30 % of atmospheric $HO_2$ was converted to OH using NO. The required NO flow, about 0.5 sccm pure NO in 10 sccm carrier flow during CYPHEX, was determined from titrations in ambient and calibration air. Further, tests were done with and without NO to ensure that there was no influence of $HO_2$ conversion into OH in the first axis. The measurement in the $HO_2$ detection cell yields the total $HO_x$ (OH + $HO_2$). $HO_2$ is calculated as the difference between $HO_x$ and the OH measured in the first axis, after accounting for the relative OH sensitivities of the two detection axes.

Due to low OH reactivity during Cyphex, we expect the $RO_2$ production from the oxidation of hydrocarbons to be low. Hence the interference due to potential conversion of atmospheric $RO_2$ radicals into $HO_2$ radicals due to the NO injected to convert $HO_2$ into OH in the low pressure detection volume is expected to be low compared to regions with high OH reactivities like the boreal forest in Finland. In order to reduce the conversion of $RO_2$ to $HO_2$, we used a reduced NO flow of 0.5 sccm resulting in ~7 x $10^{12}$ $cm^{-3}$ of NO in the detection cell, thus converting only about 30% of $HO_2$ radicals to OH radicals while simultaneously reducing the $RO_2$-$HO_2$ conversion efficiency. We did not conduct experiments to measure the conversion efficiencies of all possible $RO_2$ radicals in the atmosphere. To estimate the potential interference due to $RO_2$ radicals on our measured signal, we made model calculations using CAABA/MECCA where most of the $RO_2$ radicals from higher hydrocarbons directly form $HO_2$ radicals after reaction with NO skipping the reaction step of alkoxy radicals with $O_2$ which is slower at reduced pressure inside the instrument compared to ambient.

The model is run at ~4 hPa to see how OH and $HO_2$ radicals evolve with time in the low pressure detection volume at different NO concentrations and is validated for calibration conditions (manuscript under preparation). The $RO_2$ radicals in the model are initialized with the concentrations generated from the model run for the base case (case III in Figure 3 of the original MS) for our study while OH and $HO_2$ radicals are initialized with measured concentrations. In the Supplementary Figure 2a, we show the evolution of OH and $HO_2$ radicals inside the detection cell after injection of NO (t=0) for NO concentrations of 7.1 x $10^{12}$ and 1.71 x $10^{14}$ $cm^{-3}$ respectively. The converted OH signal is detected after 6.6 msecs (time of detection). For the high conversion efficiency case, the contribution of the $RO_2$ to the OH signal at 6.6 msecs is about 35% or 31% of the initial $RO_2$ mixing ratio. This value matches with estimates from previous study by Hens et al., 2014. For the low conversion efficiency case that represents the CYPHEX measurement mode, the estimated contribution of $RO_2$ to the measured signal is about 12 % or 2.5% of the initial $RO_2$ mixing ratio.

Further, we estimate that more than 50% of the $RO_2$ interference is due to isoprene oxidation products. This is due to the fast conversion of isoprene-based hydroxy peroxy radicals towards $HO_2$ and OH. These isoprene-based hydroxy peroxy radicals have one of the largest conversion efficiencies of up to 90 % (Fuchs et al., 2011, Lew et al., 2018). Moreover, we see that the $RO_2$ interference does not increase with increasing terpene concentrations and is nearly constant at terpene levels greater than 80 pptv (Supplementary Figure 2b), mostly because during the course of the day $HO_2$ concentration increases during this time faster than terpene based $RO_2$ concentration. This indicates that the

RO$_2$ interference effects cannot explain the deviation of modelled HO$_2$ w.r.t. measurements at high terpene mixing ratios (discussed later in sections 3.3 and 3.4).

Calibration of the instrument for OH and HO$_2$ measurements is achieved by measuring the signals generated by known amounts of OH and HO$_2$ (Martinez et al., 2010) in a calibrator setup. The calibrator was mounted on top of the OH inlet without the IPI. Known amounts of OH and HO$_2$ were produced by irradiating different concentrations of humidified air with 185 nm radiation produced by a pen ray Hg lamp. The actinic flux density of the Hg lamp (Pen-ray line source, LOT-Oriel, Germany) used for the photolytic radical production was determined before and after the campaign using the actinometry method by N$_2$O photolysis (Martinez et al., 2010). Different H$_2$O mixing ratios were produced by mixing different combinations of humidified and dry air flows using mass flow controllers. The humidified air was generated by bubbling dry air into a container half-filled with water maintained at 25-30˚C. The water mixing ratio in the humid air stream was measured by a LICOR CO$_2$/H$_2$O-Analyzer (Li-7000) based on the detection of differential absorption with an IR spectrometer. The stability of the Li-7000 was ensured by calibrations with a dew point generator (Li-610 from LICOR).

The precision and minimum detection limit for OH measurements during CYPHEX was determined from the variability of the background signals (Table 1). The chemical background, during propane injection, being larger than the spectroscopic background (off-resonance), has a dominating impact on the precision. The precision for OH and HO$_2$ measurements were calculated to be 4.8e$^5$ molecules cm$^{-3}$ and 0.39 pptv, respectively for 4 min and 14 sec time resolutions, respectively. The accuracy for OH measurements was derived from the uncertainties in OH calibrations, which involve uncertainties in determination of lamp flux (traceable to a NIST NO standard), flows (calibrated with a DC-2B, traceable to IS0/RVA 17025), H$_2$O mixing ratios (calibrated with a Li-610) and uncertainties in estimation of OH losses in the IPI, accumulating to 28.5% (2σ). The accuracy in HO$_2$ measurements was estimated to be 36 % (2σ) based on calibrations, loss of HO$_2$ in the IPI and the uncertainty in NO mixing during titrations.

2.3 Measurements of other Chemical and Meteorological parameters

During CYPHEX, measurement instruments (Table 1) were set-up in four air-conditioned laboratory containers, placed in two stacks of two. An 8 m tall, 0.5 m diameter, high flow (10 m$^3$ min$^{-1}$) common inlet installed between the stacks was used to draw ambient air for most instruments. The references to the various measurements are indicated in Table 1. C$_2$-C$_4$ alkanes and alkenes (ethane: C$_2$H$_6$, ethene: C$_2$H$_4$, propane: C$_3$H$_8$, propene: C$_3$H$_6$, butane: C$_4$H$_{10}$ (i and n), butene: C$_4$H$_8$ (c and t) were measured with a time resolution of 60 min and the mixing ratio represents an average over a sampling period of 20 min (Sobanski et al., 2016). Photolysis frequencies were obtained by a CCD Spectroradiometer (Metcon GmbH) operating at 275-640 nm with a 2-π integrating hemispheric quart dome. The spectroradiometer was calibrated prior to the campaign using a 1000 W NIST traceable irradiance standard. Photolysis frequencies were calculated using molecular parameters recommended by the IUPAC and NASA evaluation panels (Sander et al., 2011; IUPAC, 2015). An automatic weather station (Vantage Pro2; Davis Instruments Corp., Hayward, CA) was used to measure temperature, pressure, wind direction and speed, solar radiation and humidity with a time resolution of 1 min.

2.4 The CAABA/MECCA model

Due to the high reactivity of OH with a plethora of anthropogenic and biogenic hydrocarbons, the $HO_x$ chemistry of the atmosphere involves numerous reactions. To efficiently evaluate these reactions, several mechanisms of various complexity have been developed, e.g. the Regional Atmospheric Chemical Mechanism (RACM; (Stockwell et al., 1997), the Master Chemical Mechanism (MCM; http://mcm.leeds.ac.uk), the Mainz Isoprene Mechanism (MIM; (Taraborrelli et al., 2009), the Carbon Bond Mechanism (Yarwood and Rao, 2005), etc. For this study, we use the Module Efficiently Calculating the Chemistry of the Atmosphere (MECCA, Sander et al., 2011; Sander et al., 2005) , which is an atmospheric chemistry module that contains a comprehensive set of gas and aqueous phase chemical reaction mechanisms covering tropospheric and stratospheric chemistry. The gas phase chemistry in the present version contains 2664 species (including 40 dummy species to account for deposition) and 1670 reactions, including basic $O_3$, $CH_4$, $HO_x$, $NO_x$, NMHC and sulfur chemistry. Complex organic chemistry is taken from the Mainz Organic Mechanism (MOM, Taraborrelli et al., 2015). MOM is based upon the MIM3, containing new additions to an isoprene oxidation mechanism (MIM2) for regional and global atmospheric modelling. Under pristine conditions, isoprene oxidation buffers OH to a narrow range of concentrations (Taraborrelli et al., 2009). The original MIM is based on the MCM. MIM2 was further developed to improve the tropospheric chemistry under a wide range of $NO_x$ regimes, and provided a better evaluation of $NO_x$ and organic nitrogen-containing species due to a detailed representation of the alkyl and peroxy acyl nitrates as well as isoprene oxidation products, some of which can be measured by modern instrumentation e.g. MVK, methacrolein (MACR), hydroxyacetone and methyl glyoxal (Taraborrelli et al., 2009). MIM3 is a reduced version of MIM2, suitable for 3D atmospheric chemistry-transport modelling. It contains an improved isoprene oxidation mechanism related to OH recycling. It accounts for the photo-oxidation of unsaturated hydroperoxy-aldehydes, a product of isoprene oxidation; initiating a hydroxyl radical production cascade. Compared to RACM, MIM3 presents a lower degree of lumping and reproduces the mixing ratios of the main chemical species in the atmosphere and many intermediates very well under a tropical mid-$NO_x$ scenario (Taraborrelli et al., 2012). In MIM3, the extended chemistry following the OH and $O_3$ pathways has been thoroughly revised, tested in a box model, and shown to reproduce $HO_x$ measurements, even for challenging conditions with high isoprene mixing ratios (>1 nmol mol$^{-1}$) and low NO (< 40 pmol mol$^{-1}$)(Taraborrelli et al., 2012). MOM has been first presented and used in the study by Lelieveld et al. (2016). It represents the gas-phase oxidation of more than 40 primarily emitted VOCs. The structure and the construction methodology mirrors the one of the MCM. The oxidation mechanism for aromatics has been presented in Cabrera-Perez et al., (2016). The terpene oxidation scheme includes previous developments for modelling HOx field measurements (Taraborrelli et al. 2012; Hens et al. 2014; Nolscher et al. 2014). Most of the known and/or proposed HOx-recycling mechanisms under low-NO conditions are taken into account. Finally, isoprene chemistry follows to a large extent Peeters et al. (2014) and Jenkin et al. (2015) with modifications by Nolscher et al. (2014). Chemistry of the pinenes (monoterpenes) is a reduction of the MCM with modifications proposed in the past by Vereecken et al., (2007); Nguyen et al., (2009); Vereecken and Peeters, (2012); Capouet et al., (2008). For the numerical integration, MECCA uses the KPP software (Sandu and Sander, 2006). All gas-phase reactions are contained in a single chemical file (gas.eqn).

The most common method to efficiently evaluate a complex chemical environment is to use a zero dimensional, photochemical box model. Here, we use version 3.8 of the box model CAABA (Chemistry As A Boxmodel Application) which is based on version 3.0 described by (Sander et al., 2011). To apply the MECCA chemistry to atmospheric conditions, it is connected to the CAABA base model via the MESSy interface. The model is constrained

by observed concentrations of the main reactive trace gases, $O_3$, CO, NO, $NO_2$, nitrous acid (HONO), HCHO, hydrogen peroxide ($H_2O_2$), methyl hydroperoxide (MHP, $CH_3OOH$), $SO_2$, $C_2$-$C_4$ alkanes and alkenes, isoprene, pinene (α and β), limonene, benzene, toluene, methane, methanol, acetonitrile, acetaldehyde, acetone, acetic acid, as well as photolysis rate constants. Model results were obtained by letting the model run into steady state for OH and $HO_2$ for each set of data points. The steady state was set to be achieved for OH when the relative changes in OH was less than

$5 \times 10^{-7}$. Deposition velocities used in our scheme for some important species are provided in Table 2. The deposition is incorporated into the model scheme by converting a species into a dummy species according to its deposition velocity. As these dummy species will not participate in any further chemical reaction, their precursors are effectively removed from the model chemical scheme. The deposition velocities of peroxides, formic and nitric acid are based on (Nguyen et al., 2015) while those for PINAL and PAN are based on (Evans et al., 2000) using the average of the

deposition rates for water, forest and grass. The model scheme is included in the supplementary of this manuscript.

2.5 The FlexPart model

In order to understand the impact of different emission sources on the atmospheric processing of air masses, the dynamical transport history of air parcels reaching Cyprus during the CYPHEX campaign was traced using FlexPart 9.2 (Stohl et al., 2005). FlexPart is a Lagrangian particle dispersion model that describes the transport and diffusion

of tracers by computing the trajectories of large number (ca. 10000) of infinitesimally small tracer particles. For CYPHEX, trajectory simulations were done at 3-hour time intervals during 21-31 July 2014. For this, FlexPart was run 120 hours backward in time from the measurement site driven with analyses from the ECMWF with 0.2° x 0.2° horizontal resolution and a temporal resolution of 1 hour (Huser et al., 2017). The particle density distribution of the tracer particles during this 120 hour backward simulation provided information on their residence time in each grid

cell of the defined geographical area (Supplementary Figure 3). Major transport routes of air reaching the site during CYPHEX were identified from the grid cells with higher residence times.

**3. Results and discussion**

3.1 $HO_x$ measurements during CYPHEX and associated meteorological and chemical parameters

The variation of OH and $HO_2$ along with a few important chemical and radiation parameters relevant to $HO_x$ chemistry

during 21 July-1 August (Day of the year: 202-213) is shown in Figure 2. The mean values of OH and $HO_2$ during 21 July-1 August were $2.2 \times 10^6$ and $2.87 \times 10^8$ molecules per $cm^3$ respectively. The mean OH concentration during peak noon hours ($J(O^1D)>2e^{-5}$ $s^{-1}$) was $5.75 \times 10^6$ molecules per $cm^3$. While both OH and $HO_2$ showed a clear diurnal variation, the $HO_2$-OH ratio decreased with enhancement in $J(O^1D)$, and was close to 100 at $J(O^1D)=3e^{-5}$ $s^{-1}$. During the MINOS campaign in the summer of 2001 in Crete, an island in the Central Mediterranean at similar latitude about

700 km west of the CYPHEX site, average OH levels were measured to be $3.6$-$6.7 \times 10^6$ molecules per $cm^3$

(Berresheim et al., 2003). The peak OH levels during MINOS reached twice the peak values observed during our study in Cyprus. During MINOS, OH peaked at about $2.1 \times 10^7$ molecules per $cm^3$, with $O_3$ and $J(O^1D)$ of about 60 ppbv and $2.8e^{-5}$ $s^{-1}$. These $O_3$ and $J(O^1D)$ values are not very different from CYPHEX values, indicating that the peak primary OH production would not be very different for MINOS and Cyprus. Although $HO_2$ was not measured during MINOS, the CO levels (mean values reaching close to 160 ppbv) were much higher (Heland et al., 2003) compared to CYPHEX indicating possibility of higher $HO_2$. Nevertheless, the absence of $HO_2$ measurements during MINOS impairs our ability to directly compare the OH chemistry from the two datasets for a regional perspective. However, the peak OH levels observed during CYPHEX are comparable to the midday OH concentrations predicted for the Finokalia Aerosol Measurement Experiments (FAME‑08) at a remote coastal site on the island of Crete, Greece (Hildebrandt et al., 2010). The FAME measurements showed the presence of aged, highly oxygenated organic aerosols (OA) during summer compared to winter despite being heavily influenced by continental air masses (53 %) during summer and aged marine air masses (61 %) during winter. The presence of highly oxygenated OA during summer were attributed to strong photochemical impacts due to higher $O_3$ and UV and levels during summer, leading to double the values of midday OH concentrations during summer compared to winter (Hildebrandt et al., 2010).

While OH measurements during CYPHEX started on 12 July, 2014 (day number: 193), systematic $HO_2$ measurements started much later from 21 July, 2014 (day number: 202). Simultaneous measurements of OH and $HO_2$ during CYPHEX are available for the period 21-31 July, and the discussion in this work is based on this common dataset. Large day-to-day changes in concentrations of several OH and $HO_2$ precursors were observed during the study period e.g. $O_3$ varied from 50-110 ppbv, CO from 70-140 ppbv, HCHO from 0.2-2 ppbv. The noontime OH and $HO_2$ levels were $5.75 \times 10^6$ ($\pm 43\%$) and $6.25 \times 10^8$ ($\pm 30\%$) molecules per $cm^{-3}$ respectively (Fig. 2). While the average concentrations of $O_3$ and CO during the study period (Day Number: 202-212) were 69 and 104 ppbv respectively, during the days 205, 206, the mean $O_3$ and CO values were 22 and 33 % lower. The later period i.e. days 205 and 206 correspond to air masses arriving from South-Western Europe and passing over the Mediterranean with predominantly marine influence (reaching over 70 % for 5 day simulations using FlexPart). The large changes in $O_3$ and CO under different air mass regimes do not translate into the changes in $HO_x$ levels (Figure 2). This is because their concentrations are to some extent buffered as sources and sinks tend to increase in parallel in relation to anthropogenic/biogenic emissions. To study the impact of these numerous reactions on the $HO_x$ budget, we used a photochemical box model (CAABA/MECCA).

3.2 Modelling simulations of CYPHEX $HO_x$

Figure 3 shows simulations of OH and $HO_2$ using CAABA/MECCA in comparison with the measurements in ambient air. Due to remote location of the measurement site, it is expected that the $HO_x$ chemistry would be representative for the background Mediterranean atmosphere. In general, NO levels were low during CYPHEX, with only ~2 % of data above 1 ppbv NO and ~5 % above 100 pptv NO. The study is limited to NO below 100 pptv to exclude measurements affected by emissions from the local diesel generator and local automobile traffic. Under very remote conditions, the steady state of $HO_x$ would be achieved from a balance between $HO_x$ production (R1-R3) and $HO_x$ loss (R10). However, such a simplified condition is rarely achieved in the boundary layer as the presence of hydrocarbons leads

to additional production and loss channels. A first model run was done constraining only $O_3$, CO, $CH_4$, NO, $NO_2$, HONO, HCHO and $H_2O_2$, $H_2O$ and photolysis frequencies to measured values (Figure 4 case I). Thus, the major primary production channels for OH and $HO_2$ via photolysis of $O_3$, HONO and HCHO are constrained by measured quantities, while the recycling of OH and $HO_2$ occurs via reactions with $O_3$, CO, $CH_4$ and NO only. However, in the absence of the contribution from important hydrocarbons, OH loss is underestimated leading to an over-prediction of OH in the order of 88 %. Further, $HO_2$ is also overestimated, on average, by more than 30 % in this case (Figure 4 case I) as loss via formation of hydroperoxides is under represented due to the incomplete VOC representation leading to too low $RO_2$ concentrations.

When we constrain the model with all measured species, the OH simulations are much improved with a model to measured ratio of 0.90 (Figure 3 case III). However, $HO_2$ is now underpredicted by about 17 %. The underprediction of $HO_2$ is enhanced with increasing terpene (α-pinene, β-pinene and limonene) mixing ratios, with average $HO_2$ model/measurement ratio of 0.72 for >80 pptv of measured terpenes. When terpenes are not included in the model, the $HO_2$ simulations are much improved (model/measured=1.02). The mean daytime OH reactivity calculated for the measured major terpenes was about 0.06 $s^{-1}$. This value is only 3.7 % of the calculated reactivity from the total of all measured species and about 10.7% of the OH reactivity ($k_{OH}$; Eq. 1) from CO. The OH reactivity calculated from the trace gases measured during CYPHEX varied between 1-2 $s^{-1}$ during the study period (Figure 4), which is comparable to reactivity measurements made in the free troposphere (Mao et al., 2009). During the TORCH-2 campaign in Weybourne, England and the DOMINO campaign in El Arenosillo, Spain, the total reactivity was generally <5 $s^{-1}$ when air masses originated from the sea (Lee et al., 2009; Sinha et al., 2012). The OH reactivity measured in suburban regions is generally higher than 5 $s^{-1}$ (Yang et al., 2016). During CYPHEX, CO and $CH_4$ constituted more than half of the total calculated reactivity (Figure 4). CO, $CH_4$, HCHO, $C_5H_8$ and $CH_3CHO$ accounted for 35, 17, 13, 9 and 8 % respectively of the calculated OH reactivity for daytime values during CYPHEX.

$$k_{OH}= \sum(k_{species} \; x \; C_{species}) \hspace{4cm} (Eq \; 1)$$

where $k_{species}$ is the rate coefficient for reaction of a given species with OH and $C_{species}$ is the concentration of the same species.

As limonene chemistry is not included in the current model scheme, it is accounted for in the model in form of additional α-pinene after normalizing for their OH reactivities. When limonene is included as $CH_4$ instead of α-pinene, the OH loss increases by about 1 % compared to case II when there were no terpenes in the model, which confirms the average OH reactivity of limonene at 0.02 $s^{-1}$ i.e. an additional 1 % of the calculated reactivity from the measured species. Although limonene chemistry is different from α-pinene in that it caters more towards secondary organic aerosols (SOA) formation, addition of limonene as α-pinene increases $HO_2$ and OH losses by 3-6 %, which is well within the uncertainty of the measurements. Nevertheless, the degradation products from terpene oxidation (shown in Figure 5) further act as sources and sinks for OH and $HO_2$, hence play an important role in the overall $HO_x$ chemistry (Calogirou et al., 1999; Librando and Tringali, 2005; Zhang et al., 2015; Wisthaler et al., 2001). Further, increasing/decreasing the terpene concentrations by their measured uncertainties (15 %; Table 1) decreases/increases the OH and $HO_2$ model/measured slope by 2-3 %, which is close to the errors on the regression slopes. Also, changing

other gases like CO and $C_5H_8$ by their uncertainties (Table 1) has less than 3 % impact on the model vs measured slopes for OH and $HO_2$. Nevertheless, when terpenes are included as $CH_4$, both OH are $HO_2$ simulations are much improved (Figure 3 case IV). In this case, the terpene chemistry does not play any role in OH-$HO_2$ simulations but the primary OH reactivity of terpenes is taken into account.

To access the model uncertainty, Monte-Carlo simulations are carried out by selecting 228 data points from the input data set representing at least two occurrences of concentrations i.e. 1-15, 25-75 and 85-99 percentiles of $O_3$, CO, NO, HCHO, HONO, $C_5H_8$, $C_{10}H_{16}$ and $J(O^1D)$. For each of these points, we performed 9999 Monte Carlo simulation runs. The Monte Carlo simulations are based on the method described in *Sander et al.* (2011) but additionally we varied the boundary conditions of the measured species by their $\pm1$ σ uncertainty (Table 1). The derived overall model
uncertainty at 1 σ is 17.4 % and 10.5 % for OH and $HO_2$ respectively.

Review studies show agreements between *in situ* observations of OH and $HO_2$ and box model calculations using different chemical schemes within a factor of 2 for both for urban and marine (boundary layer) environments (Stone et al., 2012). During the North Atlantic Marine Boundary Layer Experiment (NAMBLEX) field campaign in the summer of 2002 at the Mace Head Atmospheric Research Station, OH was underpredicted early in the morning and
in the late afternoon/early evening and overpredicted in the middle of the day based on simple steady state calculations (Smith et al., 2006). However, photolysis of HONO was not considered due to the absence of HONO measurements during NAMBLEX. The authors contemplate that HONO photolysis and OH recycling could play crucial roles in explaining the discrepancy between measured and calculated OH during periods with high solar zenith angle. Despite low isoprene regimes (Lewis et al., 2005), inclusion of DMS and $C_5H_8$ improved calculated/measured ratio for OH
from $1.13\pm0.36$ to $0.94\pm0.39$ during NAMBLEX. $HO_2$ was overpredicted but model performance improved by incorporating reactions with BrO, IO and aerosol uptake; the best case scenario for $HO_2$ modeled/measured was $1.26\pm0.36$. During the Intercontinental Chemical Transport Experiment-A (INTEX-A) , OH was predicted fairly well for the lowest 2 km over North America and the western Atlantic Ocean using a zero-dimensional, time-dependent photochemical box model developed at NASA Langley Research Center (Ren et al., 2008). In the study, $HO_2$ was
overpredicted (model/measurement=1.37). It is challenging to put the model-measurement agreement during CYPHEX in perspective of other studies, because they vary with respect to chemical regimes i.e. different amount of $NO_x$ and hydrocarbons as well as the chemical mechanisms employed to simulate the $HO_x$ chemistry. (Rohrer et al., 2014) compared the $HO_x$ chemistry from different field campaigns using a common modelling framework, and point to buffering mechanisms that maintains OH in various $NO_x$ and VOC regimes. (Ren et al., 2008) have noted that
despite highly constrained measurement suites during various field campaigns, there are significant discrepancies between model predictions and actual measurements of OH and $HO_2$ in different environments. One possible reason for the discrepancy could be due to unmeasured atmospheric constituents. However, during CYPHEX we have been able to measure the major primary sources and sinks related to $HO_x$ chemistry and simulations agree within uncertainty of OH and $HO_2$ measurements. Another plausible reason for discrepancy is the uncertainty in mechanistic pathways,
branching ratios and rate constants etc., which are not known with sufficient accuracy. A recent study comparing 7 different chemical mechanisms constrained with the same input data and boundary conditions revealed 25-40 %

ambiguity in predictions of $HO_2$ and OH over the United States (Knote et al., 2015). As we highlight in the next section, also the $HO_x$ chemistry related to terpenes is associated with substantial uncertainty.

3.3 Reasons for decreased model accuracy when including the terpene chemistry

As discussed in the last section, we find that the underprediction in $HO_2$ gradually increases with increasing terpene levels, being about 38 % for the highest terpene levels observed (about 120 pptv). To ascertain if this underprediction is related to terpene chemistry or some other chemical pathway, we compare two model runs viz. case III vs case IV in Figure 3. Case IV is achieved by initializing the model with zero terpenes, however keeping the terpene reactivity towards OH intact. The terpene reactivity i.e. the product of individual terpene concentrations with their rate coefficient towards OH is approximated in the model scheme in form of additional $CH_4$ (Figure 3, case IV). Under this scenario, we find that the agreement between modelled and observed $HO_2$ again improves, to levels similar to case II without terpenes in the model (Figure 3). The small difference of 6 % with respect to OH prediction between these cases II and IV (Figure 3) also corroborates the minor primary reactivity of <0.06 s$^{-1}$ for the sum of α-pinene, β-pinene and limonene towards OH. However, up to 38 % underprediction in $HO_2$ for terpene concentrations over 80 pptv indicates the important role of secondary products formed during terpene oxidation towards the $HO_x$ budget.

To quantify the magnitude of the impact of terpene degradation products in the $HO_x$ budget, we compared the $HO_x$ sinks from the two model cases: case III (Figure 3, henceforth 'terpene case') and case II (Figure 3, henceforth 'isoprene case'), the difference being the presence and absence of α-pinene, β-pinene and limonene. The estimation of $HO_x$ sinks from the two model cases is conspicuously marked by significant contribution of terpene degradation/oxidation products in the terpene case which are absent in the isoprene case. The additional reactions in the terpene case make up 6.3-27.8 % of the total $HO_x$ sinks, the contributions increasing with the terpene concentrations and peaking between 13-14 hours Local Time on clear days (Figure 5). While the terpene degradation products contribute, on average, 15 % to the total $HO_x$ loss, this value is 20.5 % for terpenes>80 pptv. The corresponding contribution of terpene related peroxy radicals to the total $HO_x$ loss is 7.8 and 10.3 % respectively, accounting for about 52 % of the impact of the terpene degradation products, including peroxy radicals, pinal, norpinal and peroxides, towards $HO_x$ loss.

Since only about 3.7 % of primary terpene reactivity with OH results in up to 28 % of $HO_x$ loss, with terpene generated peroxy radicals having more than four times the impact of the primary terpenes to the $HO_x$ loss, we examine a sequence of reactions starting with the oxidation of α-pinene to understand the genesis and propagation of these terpene generated peroxy radicals (R15-R23). The reactions show that oxidation of α-pinene via OH and $O_3$ directly generates several peroxy radicals (R15-R16) which are sinks for $HO_x$ via their reactions with $HO_2$ forming hydroperoxides (e.g. R21-R22). These first generation peroxy radicals also form other peroxy radicals (R17-R20), and the chain propagates to produce additional peroxy radicals contributing via formation of the respective hydro peroxide in reaction with $HO_2$ leading to overall $HO_x$ loss (e.g. R23).

| | | | | | |
|---|---|---|---|---|---|
|  + OH | → 0.75 | <br>$C_{10}H_{17}O_3$<br>(APINABO2) | +0.15 | $HO_2$ | +… | (R15, G40200 in Model) |
|  + $O_3$ | → .09 | <br>$C_{10}H_{17}O_3$<br>(APINABO2) | + .33 | <br>$C_9H_{14}O_2$<br>(Norpinal) | + .77 OH<br>+ .33 CO<br>+ .33 $HO_2$<br>+…. | (R16, G40232 in Model) |

| | | | | | |
|---|---|---|---|---|---|
| <br>$C_9H_{14}O_2$<br>(Norpinal) | + hv | → | <br>$C_9H_{14}O_2$<br>(Norpinenol) | | (R17, J49203b in Model) |
| <br>$C_9H_{14}O_2$<br>(Norpinal) | + hv | → | <br>$C_8H_{13}O_3$<br>(C85O2) | + CO + $HO_2$ | (R18, J49203a in Model) |
| <br>$C_9H_{14}O_2$<br>(Norpinal) | + HCOOH | → | <br>$C_9H_{14}O_2$<br>(Norpinenol) | + HCOOH | (R19, G49248 in Model) |

| | | | | | |
|---|---|---|---|---|---|
| <br>$C_9H_{14}O_2$<br>(Norpinenol) | + OH | → | <br>$C_8H_{13}O_4$<br>(C86O2) | + HCOOH<br>+ OH | (R20, G49246 in Model) |

| C$_8$H$_{13}$O$_3$ (C85O2) | + | HO$_2$ | → | C$_8$H$_{14}$O$_3$ (C85OOH) | (R21, G48201 in Model) |
|---|---|---|---|---|---|
| C$_8$H$_{13}$O$_4$ (C86O2) | + | HO$_2$ | → | C$_8$H$_{14}$O$_4$ (C86OOH) | (R22, G48206 in Model) |

| C$_8$H$_{13}$O$_4$ (C86O2) | → | C$_5$H$_7$O$_4$ (C511O2) | + CH$_3$COCH$_3$ | (R23, G48204 in Model) |
|---|---|---|---|---|

Thus, a cascade of HO$_x$ scavengers mostly in the form of RO$_2$ species, at low NO$_x$ levels, starting with oxidation of primary terpenes leads to nearly 28 % HO$_x$ loss, resulting in up to 38 % underprediction of HO$_2$. The losses increase with increases in terpene levels. The major contribution towards the HO$_x$ loss comes from terpene generated organic peroxy radicals (Figure 3). However, there are major uncertainties regarding the chemical pathways and rate coefficients governing the chemistry of these peroxy radicals in the actual atmosphere and in the next section we examine several postulates based on the literature.

3.4 Terpene generated organic peroxy radicals

A limitation in our understanding of peroxy radical oxidation pathways is the rate constant of peroxy radicals with NO. For most peroxy radicals, lumped rate coefficients are defined by the expression Eq. 2 (Rickard and Pascoe, 2009) as kinetic data are unavailable for most of the RO$_2$-NO reactions.

$$K_{RO2NO} = 2.54 \times 10^{-12} \exp(360/T).f \qquad \text{Eq 2}$$

Thus, K$_{RO2NO}$ is the product of the rate coefficient and an efficiency factor, $f$. The values of $f$ for different classes of RO$_2$ are given in (Jenkin et al., 1997). The reaction of RO$_2$ with NO can proceed via two channels, one forming alkyl nitrate (RONO$_2$) and the other forming RO and NO$_2$ (R12-R13). Enhancing the RO$_2$-NO rate coefficient suppresses the peroxide forming channel via HO$_2$. For our study period, increasing the KRO$_2$NO even by a factor of 4 led to only a 6.8 % increase in modelled HO$_2$ levels for terpenes> 80 pptv (Figure 6).

During the summertime 1999 Southern Oxidant Study (SOS99) at Nashville, Tennessee, agreement could be found between photochemical models and *in situ* measurements when the product of the branching ratio and rate constant for organic peroxide formation, via R11, was reduced by a factor of 3-12 (Thornton et al., 2002). To test this hypothesis, we decreased the $K_{RO2HO2}$ for terpenes by half and by one-fifth of its value considered in the model. The rate coefficient of $RO_2$ with $HO_2$ leading to ROOH is given by Eq. (3), where $K_{RO2HO2}$ increases with the size of $RO_2$ radicals (Atkinson et al., 1999; Boyd et al., 1996) http:// mcm.leeds.ac.uk/MCMv3.3.1/categories/ saunders-2003-4_6_4-gen-master.htt?rxnId=6942).

$$K_{RO2HO2} = 2.91 \times 10^{-13} \exp(1300/T) [1 - \exp(-0.245n)] \qquad \text{Eq 3}$$

where *n* is the carbon number and T is the temperature.

We find that decreasing the rate coefficient of by a factor of 5 leads to better reconciliation with our observations than when decreasing it by a factor of 2. When $K_{RO2HO2}$ was reduced by a factor of 5, the modelled $HO_2$ increased by only 9.9 % for higher terpene levels i.e. α-pinene+ β-pinene+limonene>80 pptv (Figure 6).

Recent studies have proposed inter and intramolecular hydrogen abstraction by peroxy radicals (autoxidation) to be effective reaction pathways leading to OH formation (Crounse et al., 2013). This abstraction is largely determined by the thermochemistry of the nascent alkyl radicals and thus is strongly influenced by neighboring substituents, hence the rate increases rapidly when more oxygen containing functional groups are involved. The H-shift rates reported in literature can be as fast as 0.1 s$^{-1}$ for atmospheric conditions. We incorporated a simple autoxidation scheme for some of the most important terpene related $RO_2$ species as specified in figure 5 in our model. In this scheme, the specific terpene generated $RO_2$ species whose contribution was at least 0.25 % to the $HO_x$ sink are converted into a corresponding ROOH at a very fast rate ($1e^{10}$ x $K_{RO2HO2}$) reaching up to 0.2 s$^{-1}$,e.g. $C_{10}H_{15}O_4$ (PINALO2) is converted to $C_{10}H_{16}O_4$ (PINALOOH). We find that this scheme is effective in increasing the model $HO_2$ levels by 24.2 % for terpene levels >80 pptv (Figure 6). Since we have used a very simple approach to test the impact of autoxidation in the model atmosphere, more studies are required towards a mechanistic development for incorporating these schemes viz. which way the H-shift will occur at which rate, which products will be formed, etc. The present exercise in implementing the autoxidation scheme is only meant to show its importance, effectiveness and potential in atmospheric chemistry models.

Further, when we greatly increased the $RO_2$-R'$O_2$ reaction rates to $4e^{-10}$ cm$^3$ molecule$^{-1}$ s$^{-1}$ (close to the collision limit), we found that the discrepancy of the modelled $HO_2$ loss w.r.t. the measurements with increasing terpenes decreases (Figure 6) and the overall agreement between modelled and measured $HO_2$ is still good (Figure 6, Supplementary Figures 4 & 5). The large deviation between observations and modelled $HO_2$ around middays on 205, 208 and 210 occur during periods with high terpene concentrations. On day 205 and 208, the simulation with the autoxidation scheme shows much better agreement to observations compared to the base case (Case III in figure 3) while on day 210, when terpene concentrations are much higher, even the autoxidation scheme fails to reproduce the $HO_2$ observations. In this case the simulation where the rate coefficient of $RO_2$-R'$O_2$ reactions was increased close to the collision limit shows a much better agreement compared to both the base case and the autoxidation case.

3.5 $HO_x$ chemistry during CYPHEX: production and loss estimates

Using the optimal scenario from our model results, i.e. the one where all measured species are included in the model (case III, Figure 3), we can study different processes impacting $HO_x$ budgets. Since impact of terpene generated $RO_2$ species was maximum on day 210 (29 July), we will ignore any special features on this day for our subsequent discussions using the model case III, the 'terpene case'. For CYPHEX, the major OH as well as $HO_x$ producing channel is the reaction of atmospheric water vapor with $O^1D$ generated from the photolysis of $O_3$ (Figure 7). Peak daytime contributions of this channel towards OH production exceeded 45 % for most of the days, and about 60 % on days 205, 208 and 209. The midday values coinciding with peak OH production on day 205 was marked by conspicuous influence of aged air masses originating over south-west Europe and considerably processed over the Mediterranean before reaching the site. The peak $HO_2$ values on this day were about 11% lower than the average peak $HO_2$ values during the study period. The peak $HO_2$ values for $J(O^1D)>2.5e^{-5}$ $s^{-1}$ was 6.4 x $10^8$ molec/$cm^3$ for the study period while this value was only 5.7 x $10^8$ molec/$cm^3$ on day 205. While, on average, the recycled OH via reactions of $HO_2$ with NO and $O_3$ (R6-R7) contributed 33.6 % to the total OH production, this value was about 6 % lower for days 205-206. It may also be noted that although $O_3$ was very low on both days 205 and 206, with predominant influence of aged marine air, the contribution from $O^1D+H_2O$ to the total OH production still exceeded 50 %. Due to lower HONO mixing ratios, the fractional contributions of HONO photolysis towards peak OH production during midday on day 208 was significantly low at about 2.5-3.5 %. On all other days, for which values are available, this channel contributed more than 6 % to peak OH production during noon time. The photolysis of HONO has the largest fractional contribution to the early morning OH production on day 211, reaching above 30 %. Overall during CYPHEX, the average daytime ($JO^1D>0$) contribution to OH production from $O_3$ photolysis and subsequent reaction of $O^1D$ with water vapor was about 39.1 %, the average daytime contribution from HONO photolysis was 12.3 %, while recycled OH from reaction of $HO_2$ with $O_3$ and NO account for 15.2 and 18.4 % of the total OH production, respectively. The four major OH producing channels (Figure 7) contribute up to 95 % of daytime OH production on most occasions, with 85.3 % on average. In addition, the reactions of acyl peroxy radicals ($RCO_3$) with $HO_2$ contribute about 3.1 % to the OH production which the photolysis of $H_2O_2$ and the ozonolysis of pinene contribute 1.85 % and 2 % respectively.

The single major sink of OH during CYPHEX was CO, followed by $CH_4$, HCHO, $C_5H_8$, $CH_3CHO$ and $O_3$, on average accounting 20.9, 10.0, 7.8, 5.1, 4.9 and 4.1 % of OH losses respectively. The reactions of various peroxides with OH to form peroxy radicals e.g.R24 contribute 8.4 % to the OH loss. Further, oxidation of $CH_3OH$, pinal, C2-C4 alkenes, α-pinene, $HO_2$, $NO_2$, $CH_3O_2$ by OH contributed 2.7, 2.4, 2.4, 2.2, 1.6, 1.4, 1.4 % respectively. During these days (205-206), the modelled OH loss (Figure 7) as well as the calculated OH reactivity (Figure 4) were lowest of the study period. It is likely that the air masses arriving to the site were already much processed, spending considerable time over the Atlantic and Mediterranean, leading to depleted OH reactivity. During this period of marine influence, the contribution of long-lived gases to the daytime OH loss increased by about 15 % while the contribution of shorter lived gases like HCHO and $CH_3CHO$ decreased by 34 % and 39 % respectively. HCHO concentrations are observed to be lowest between days 205-206, when the marine influence was the highest. The average lifetime of HCHO for

the study period is calculated to be 0.86 days, and its concentrations are higher whenever air masses are influenced by South-East Europe/ Black Sea region.

| $C_8H_{13}O_4$ (C86OOH) | + OH | → | $C_8H_{13}O_4$ (C86O2) | (R24, G48207 in Model) |
|---|---|---|---|---|

The major source of $HO_2$ during CYPHEX was the oxidation of CO by OH, contributing on average, 36.2 % of daytime $HO_2$ production values. On days 205, 206, when the site experienced aged oceanic air masses and CO concentrations were the lowest of the study period, yet the peak contribution of CO to daytime $HO_2$ production was about 40.4 %. This shows that the contribution from other $HO_2$ precursors like HCHO decreases much faster than CO because of their shorter lifetime as the air mass passes over the ocean, and isolated from their emission sources. The contributions of HCHO towards $HO_2$ production varied between 8-23 % with an average daytime contribution of 15.6 % and included both oxidation via OH and photolysis. The methoxy radical ($CH_3O$), formed during oxidation of $CH_4$ and $CH_3O_2$, was also a significant source of $HO_2$ with peak contributions reaching 23 % on some occasions and an average daytime contribution of 17.8 %.  On average, the reactions of $HO_2$ with NO and $O_3$, recycling OH, contributed 23.3 and 18.2 % respectively towards daytime loss for $HO_2$. $HO_2$ losses by its self-reactions and reactions with $CH_3O_2$ reached peak daytime values of over 40 % and on average accounted for 30.3 % of daytime $HO_2$ losses. Among these, the major contribution was from the $HO_2$ self-reactions, with peak values of over 30 % during most days, and average values of 22.4 %. The reactions of peroxy radicals ($RO_2$ and $RCO_3$) with $HO_2$ contribute 24.6 % to the $HO_2$ radical loss, resulting in increased underprediction of $HO_2$ by the model with increasing terpene concentrations (Supplementary Figure 6). $HO_2$ losses due to recycling via NO and $O_3$ and the $HO_2$ losses via reactions with itself and $CH_3O_2$, are complementary during a diurnal cycle. The self-reactions gain in importance and peak around midday as the peroxy radical concentrations increase with increasing photochemistry, while recycling reactions dominate during periods with larger solar zenith angle. During CYPHEX, for small solar zenith angles with $J(O^1D)>2e^{-5}$ $s^{-1}$, the contribution of recycling and self-reactions to $HO_2$ loss was 36.2 and 32.7 % respectively. Overall, it is found that lowest $HO_x$ production and losses occurred in the highly processed air masses influenced by the marine boundary layer, mostly over the Mediterranean Sea, but sometimes as far as the Atlantic ocean (days 205-206) with some influence from south-west Europe/northern Africa (Figure 7, panel 3).

The close proximity of Cyprus to several countries with different socio-economic conditions makes its air quality vulnerable to the increases or decreases of primary and secondary air pollutants in one of these countries in Europe, Asia and Africa. For example, increased $NO_x$ emissions in one of the Southern European countries or high CO from forest fires in Ukraine can be quickly transported to Cyprus in a matter of hours to days and modulate the atmospheric processing over the region. Increased $NO_x$ emissions can impact the $O_3$-$NO_x$-VOC chemistry by changing the $HO_x$

cycling. Even though the study period was characterized by NO mostly less than 100 pptv, decreasing $NO_x$ by 30 % still decreased OH levels by 14%, these impacts being higher in the early morning and late afternoon. On the other hand, increasing NO by 5 and 10 times caused 44 % and 70 % average enhancement in OH levels.

Because of its important role in $O_3$ production, $NO_x$ emissions have been the focus of air quality investigations during preceding decades. Apart from their crucial role in $O_3$ production, $NO_x$ levels also play a significant role in the self-cleaning capacity of the atmosphere due to their reactions with atmospheric $HO_2$ and $RO_2$. We test the impact of mainland Europe $NO_x$ emissions on the self-cleaning capability of the climatically vulnerable Mediterranean atmosphere using the CYPHEX dataset. The recycling efficiency for OH (REf-OH) is defined as the ratio of OH produced from secondary sources via reactions of $HO_2$ with NO and $O_3$ (R6-R7) to the OH produced from primary and secondary sources. The primary sources of OH considered in our calculation include reaction of $H_2O$ with $O^1D$ produced from $O_3$ photolysis and the photolysis of HONO, $H_2O_2$ and organic peroxides. The contribution of the primary OH production from HONO photolysis is corrected by subtracting off the contribution from the recombination reaction of OH with NO.

While the present study is limited to NO <100 pptv, modelling results show that REf-OH increases with increase in NO levels (Figure 8). The REf-OH increases from 0.28 at 10 pptv to 0.42 at around 30 pptv NO to 0.7 at 100 pptv NO (Figure 8). The REf-OH of 0.7 corresponds to a chain length of 2.8. The relationship of REf-OH with NO indicates that at higher NO (not observed for the CYPHEX data), the REf-OH would increase further. To understand if REf-OH would reach runaway conditions or saturate at a particular value, we chose several points along the REf-OH-NO curve and for each of these points, we generated hypothetical input data set keeping all parameters constant but changing the $NO_x$ levels from 0.2 to 3 ppbv, while maintaining the original $NO/NO_2$ ratio (Figure 8).

Figure 8 shows that in general the REf-OH increases to about 0.85 at 500 pptv NO. After this, it decreases gradually, related to the reaction of $NO_2$ with OH. Concurrently, the loss of $HO_2$ due to self-reactions and reaction of $HO_2$ with $CH_3O_2$ becomes much weaker compared to the loss of $HO_2$ to NO with increasing NO levels. The ratio of $HO_2$ losses via the radical-radical interactions (i.e. the reactions of $HO_2$ with $HO_2$ and $CH_3O_2$) to the $HO_2$ losses via the recycling channel (i.e. reactions of $HO_2$ with NO and $O_3$) decreases exponentially becoming less than 1 at 35 pptv NO and less than 0.5 at 70 pptv NO and 0.01 at 1 ppbv NO. The secondary OH production also shows a similar pattern with respect to REf-OH, peaking around 500 pptv for the different input data sets and decaying thereafter. The model generated OH peaks in this range but the $HO_x$ concentrations start to drop much earlier, from around 70 pptv NO. At these low NO levels, $HO_x$ loss by peroxide-peroxide reactions is comparable to the channels recycling $HO_2$ into OH, which can then be removed through reaction e.g. with $NO_2$. However, the recycling efficiency is only about 0.6 at 70 pptv NO. Budget analysis indicates that $HO_x$ production drops with increasing NO levels for this test dataset constrained to specific $NO_x$ levels. Overall, for the hydrocarbon levels observed during CYPHEX, the self-cleaning capacity of the atmosphere peaks at around 500 pptv NO, which at a $NO/NO_2$ ratio of 0.2 is equivalent to about 3.0 ppbv $NO_x$. Increasing $NO_x$ emissions further would lead to decreased $HO_x$ levels and the recycling efficiency is unlikely to increase further. On the other hand, decreasing NO levels below a few hundred pptv of NO would also decrease the

recycling efficiency for the same hydrocarbon levels indicating that small amount of $NO_x$ help sustain the self-cleaning capacity of the atmosphere (Figure 8).

### 4. Conclusions

Atmospheric OH and $HO_2$ were measured as part of a comprehensive atmospheric chemistry field measurement campaign conducted in Cyprus in the summer of 2014 to study the major processes impacting atmospheric oxidation and air chemistry in a relatively unpolluted coastal region, periodically influenced by long range transport of European emissions. A comprehensive suite of atmospheric chemistry measurements obtained during CYPHEX enabled a detailed investigation of atmospheric oxidation processes, using a photochemical box model, under low $NO_x$ conditions. The box model (CAABA/MECCA with MOM chemistry) simulations for OH agreed to within 10 % with *in situ* OH observations. Model simulations for $HO_2$ agreed to within 17 % of the *in situ* observations. However, the model strongly underpredicted $HO_2$ at high terpene (α- and β-pinene, limonene) concentrations (>80 pptv), this underprediction reaching up to 38 % at the highest terpene levels due to loss of $HO_2$ by reactions with terpene generated peroxy radicals. Comparison of alternative reaction pathways to reduce this unrealistic $HO_2$ loss showed that autoxidation can be an effective sink for the $RO_2$ species generated from terpene oxidation. However, low terpene and NO regimes, prevailing during CYPHEX, and the absence of limonene chemistry in the current model scheme, precluded a more rigorous analysis of the probable chemical pathways for $RO_2$ degradation in the atmosphere. Further, there is evidence that the rate constant of $CH_3O_2$ with OH could be two times faster than used in the current models (Bossolasco et al., 2014). Applying this in general to other peroxides, the rate constants need to be revisited as they will have a non-negligible impact on the chemical composition of the atmosphere, especially in remote low $NO_x$ environments where the peroxide lifetimes are relatively long. As already indicated in literature, there is large uncertainty in the rate coefficients of different $RO_2$ species with NO and $HO_2$ (King et al., 2001) which are scopes for future studies.

The radical budget analysis for CYPHEX showed that primary production of OH via photolysis of $O_3$ constituted the main OH source and peak daytime contributions from this channel exceeded 45 % for most of the days. The average daytime contribution from HONO photolysis was 12.3 % but it exceeded 30 % in the early morning on a few occasions. The recycled OH from reaction of $HO_2$ with $O_3$ and NO accounted for 14.5 and 17.7 % of the total OH production, respectively. The maximum observed recycling efficiency was about 0.7 for about 100 pptv NO. CO was not only the single major daytime sink of OH, accounting for nearly one-fifth of OH loss during the study period, but also the single major $HO_2$ production source. Despite low $NO_x$ regimes during CYPHEX, NO contributed more than 23 % of peak daytime loss for $HO_2$, reaching over 50 % on a few occasions. Lowest $HO_x$ production and losses occurred in the highly processed air masses with low OH reactivity i.e. low precursor levels. These air masses were heavily influenced by the marine boundary layer, mostly by the Mediterranean Sea, but sometimes as far as the Atlantic Ocean (24-25 July, DOY 205-206) with some influence from south-west Europe/northern Africa. Additionally, our results connote the need for deeper understanding of the reaction channels for organic peroxides in the atmosphere.

### Data availability

Data are available on request from the CYPHEX archive at MPIC server. If desired, please send an email to hartwig.harder@mpic.de.

**Competing interests**

The authors declare that they have no conflict of interest.

5 **Acknowledgement**

We acknowledge the contribution of the entire CYPHEX team, especially those members whose names are not listed as co-authors, for the success of the CYPHEX campaign. We also acknowledge the administrative and logistic support provided by the Cyprus Institute and the officials at the army base at Inea. We sincerely acknowledge the technical support provided by the workshop and electronics department in the Max Planck Institute for Chemistry in preparation

10 for the campaign. Further, technical help from Markus Rudolf, Korbinian Hens, Cheryl Ernest and Pippa Jones during preparation stage of the campaign is highly appreciated. We thank the handling editor and the two anonymous reviewers for their constructive comments and suggestions that has greatly improved the quality of this manuscript.

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

## 5. Figures

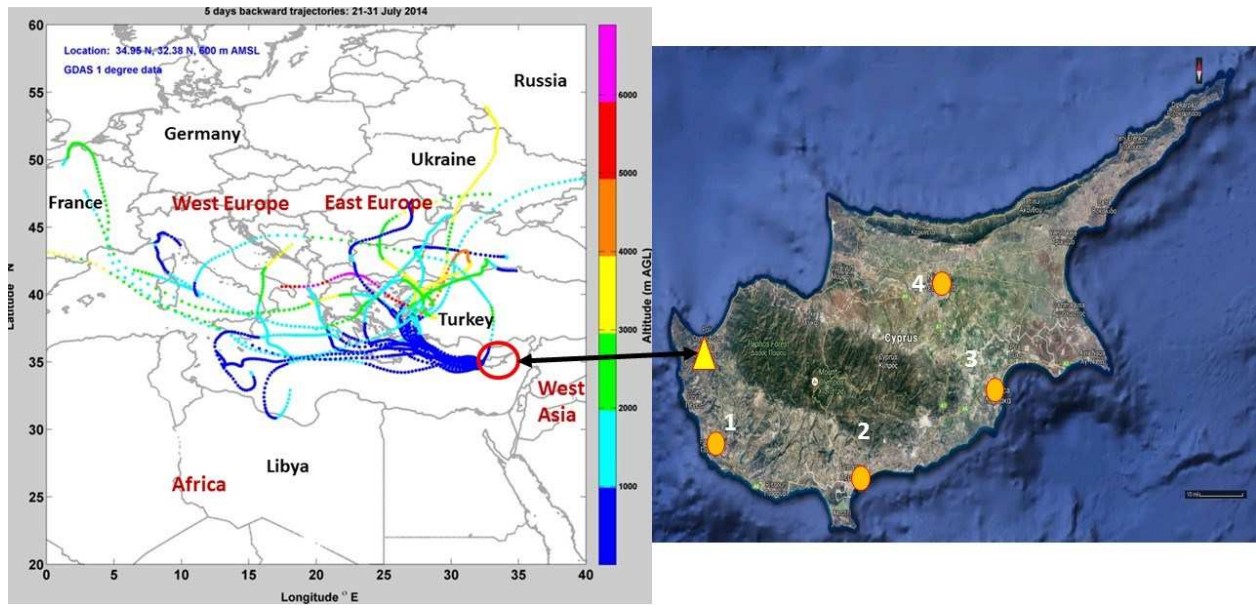

**Figure 1.** *Left: Air masses during CYPHEX; Right: Map of Cyprus. Measurement site (yellow triangle), cities (circles, 1.Paphos, 2. Limassol 3. Larnaca 4. Nicosia)*

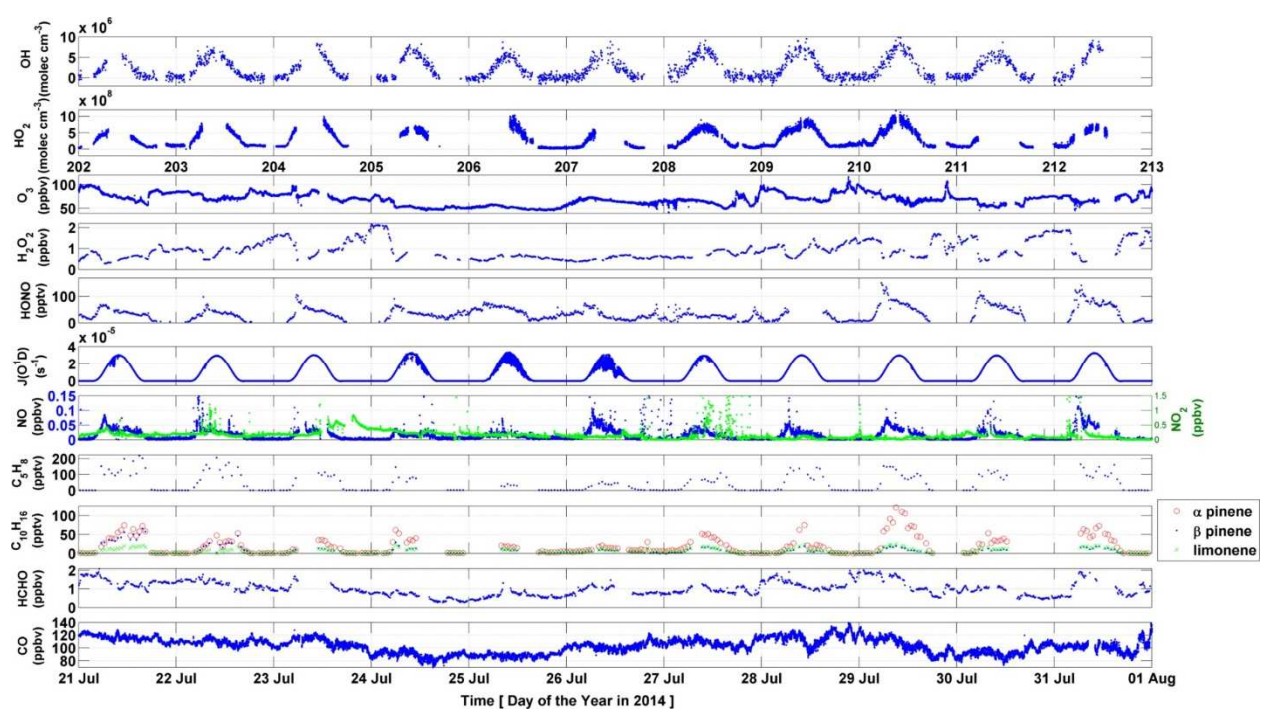

**Figure 2.** *Measurements of OH, HO₂ along with selected chemical and radiation parameters relevant to HOₓ chemistry during the CYPHEX campaign. Time is in UTC. Local time in Cyprus during summer is UTC+3.*

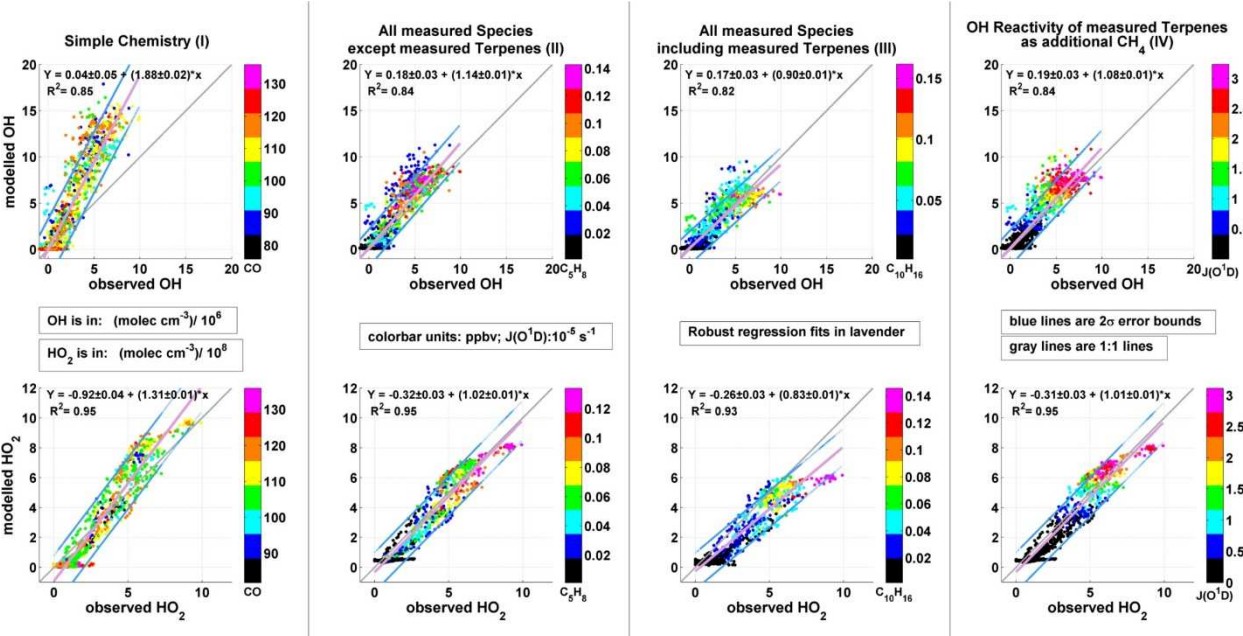

**Figure 3.** *LIF-FAGE measurements of OH (top panel) and HO$_2$ (bottom panel) vs model simulations with CAABA/MECCA. From left to right: model simulations using simple chemistry without NMHCs (case I), simulations including anthropogenic and biogenic hydrocarbons except terpenes (case II), after adding terpenes(case III) and after initializing with zero terpenes but with CH$_4$ increased to account for the additional measured terpene reactivity (case IV).*

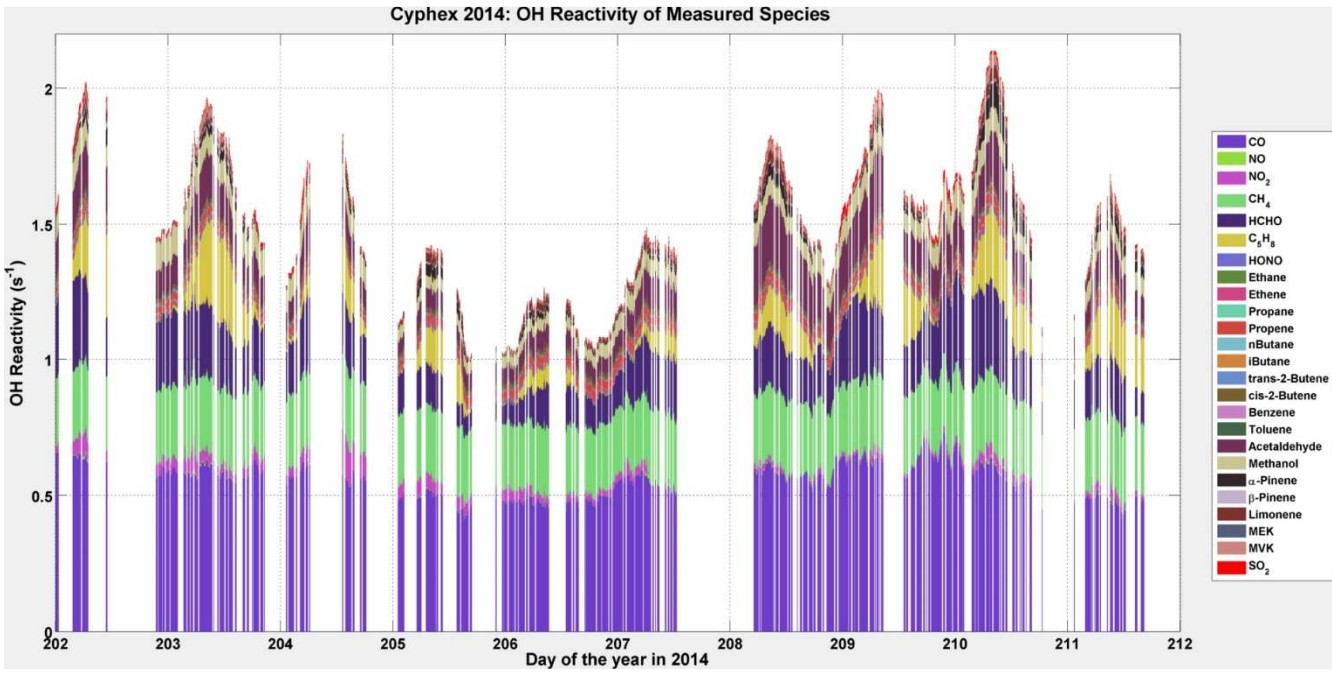

**Figure 4.** *Calculated OH reactivity of various chemical species measured during CYPHEX.*

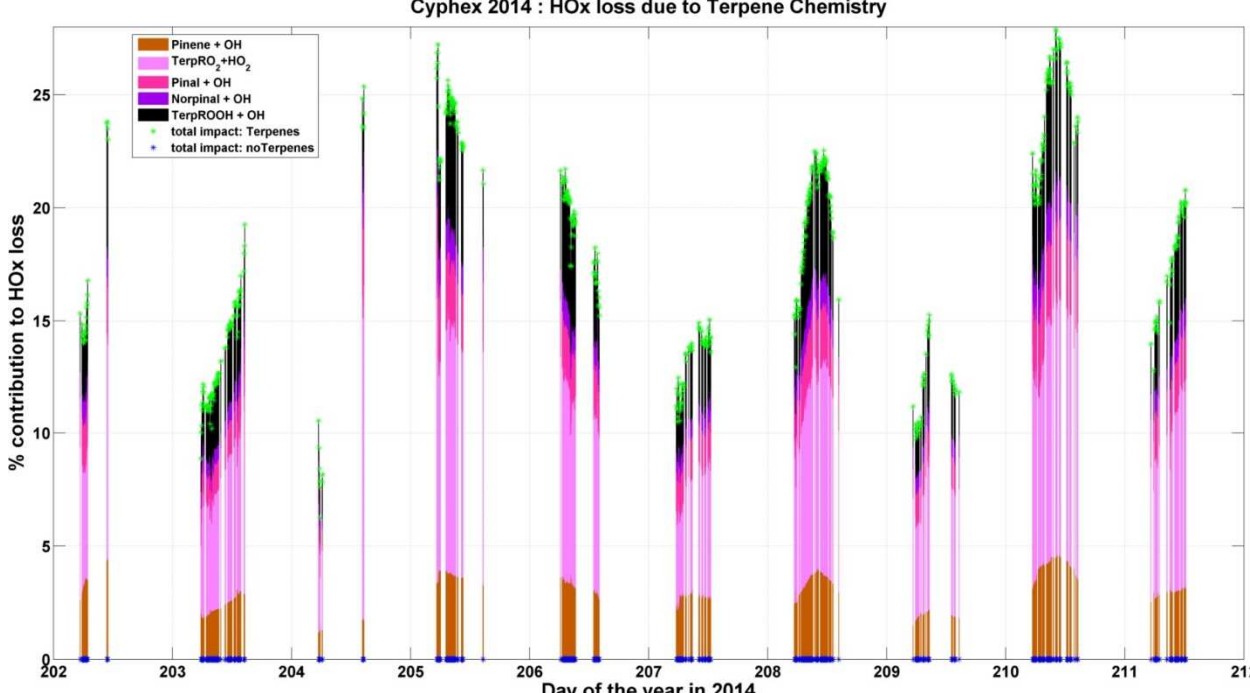

**Figure 5.** *Contribution of different terpene related reactions to the total $HO_x$ loss. Additional peroxy radicals are produced in the model by terpenes rather than organics e.g. isoprene degradation. Pinene includes α-pinene and β-pinene. TerpRO2 include only $C_8H_{13}O_3$ (C85O2), $C_8H_{13}O_4$ (C86O2), $C_9H_{15}O_3$ (C96O2), $C_9H_{15}O_4$ (C97O2), $C_9H_{15}O_5$ (C98O2), $C_{10}H_{15}O_5$ (C106O2), $C_5H_7O_4$ (C511O2), $C_6H_9O_5$ (C614O2), $C_{10}H_{15}O_4$ (PinalO2) and $C_{10}H_{17}O_3$ (BPinaO2).TerpROOH include only $C_{10}H_{16}O_4$ (PinalOOH), $C_8H_{14}O_3$ (C85OOH), $C_8H_{14}O_4$ (C86OOH), $C_9H_{16}O_4$ (C97OOH), $C_9H_{16}O_5$ (C98OOH), $C_{10}H_{16}O_5$ (C106OOH) and $C_5H_8O_4$ (C511OOH). The names in brackets are MCM nomenclature. Only those species are selected whose contribution is at least 0.25 % to the $HO_x$ sink.*

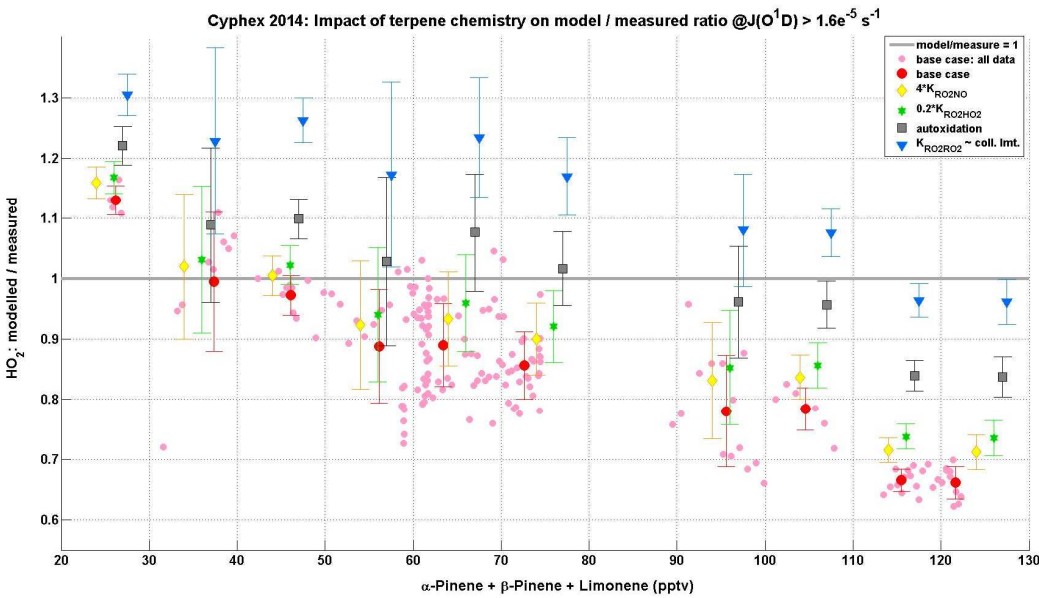

**Figure 6.** *HO₂ predictions using different chemical schemes which were employed to investigate the impact of peroxy radicals as HO₂ sink. The x-axis shows the sum of measured α-pinene, β-pinene and limonene. The base case is the model simulation using measured species; 4\*KRO₂NO is a simulation where the rates of selected terpene-related RO₂ species with NO are increased by a factor of 4; 0.2\*KRO₂HO₂ is a simulation where the rates of these RO₂ species with HO₂ are reduced by a factor of 0.2; the autoxidation simulation includes an autoxidation scheme rapidly converting selected RO₂ species into ROOH. The last scheme refers to a simulation where KRO₂R'O₂ is set to 4e^{-10} cm³ molecule^{-1} s^{-1} (close to the collision limit). The mean and sigma are calculated for 100 pptv bins along the x-axis.*

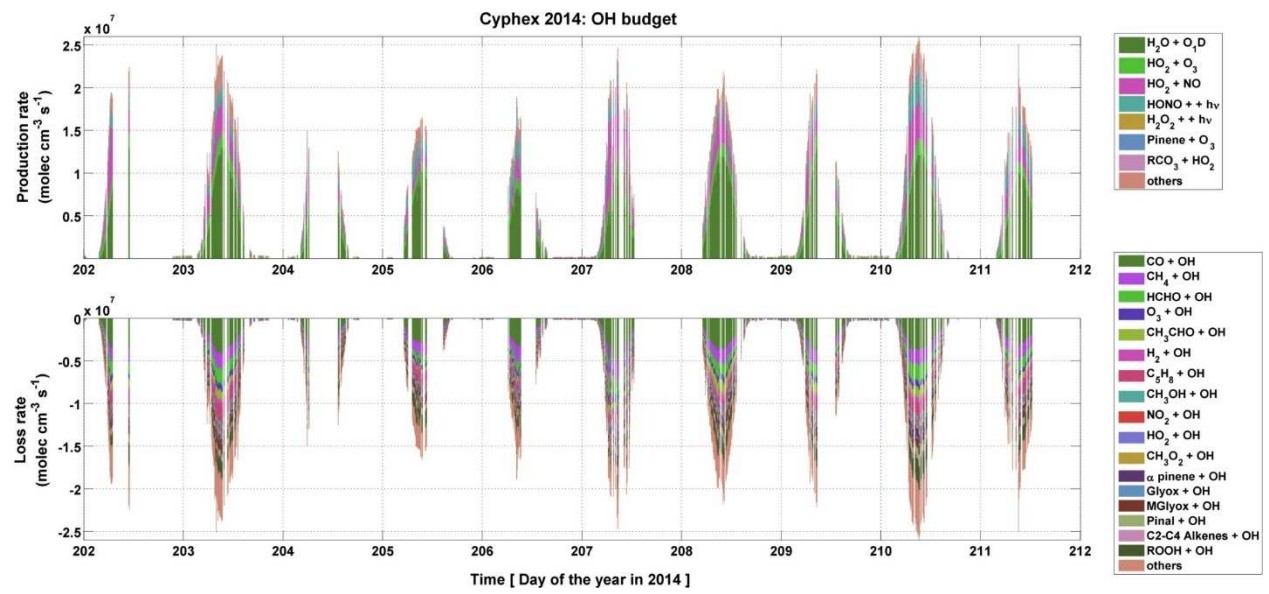

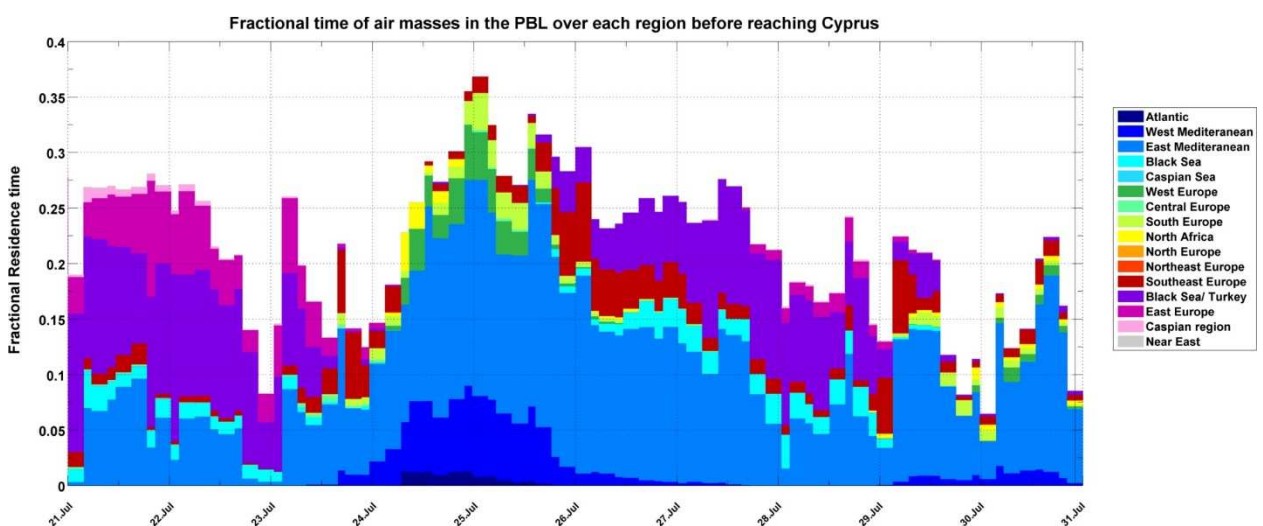

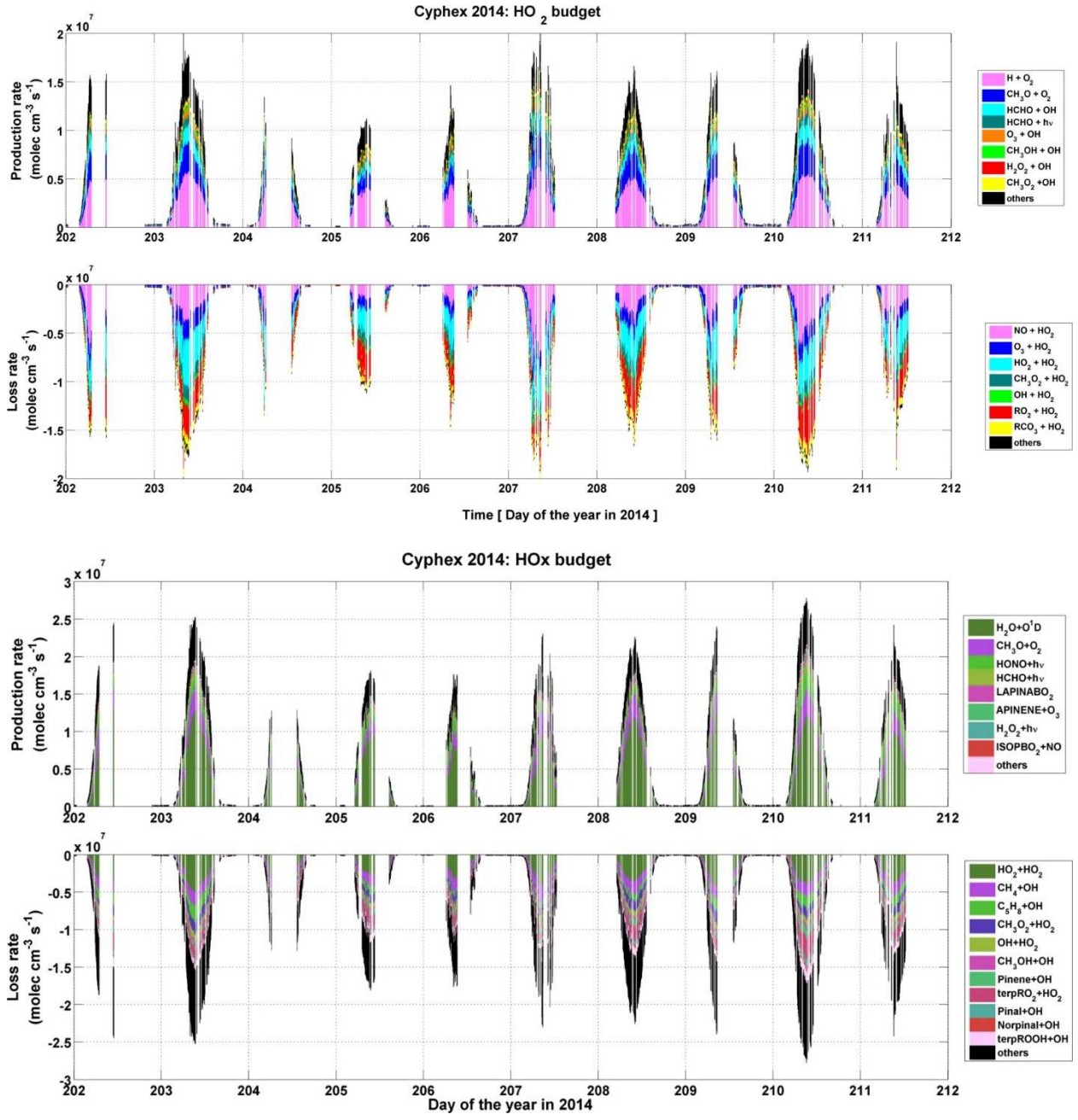

**Figure 7. a.** *Budget analysis for OH during CYPHEX. The HO$_x$ budget is significantly impacted by the changes in air mass source regions.* **b.** *Flexpart evaluation of the contribution of different source regions within the planetary boundary layer to the air mass origin at the measurement site in Cyprus is shown in Panel 3. The air mass source regions are shown in a separate colour map (supplementary Figure 1).* **c.** *Budget analysis for HO$_2$,* **d.** *Budget analysis for HO$_x$, during CYPHEX. The descriptions for Pinene, terpRO$_2$ and terpROOH is same as in Figure 5.*

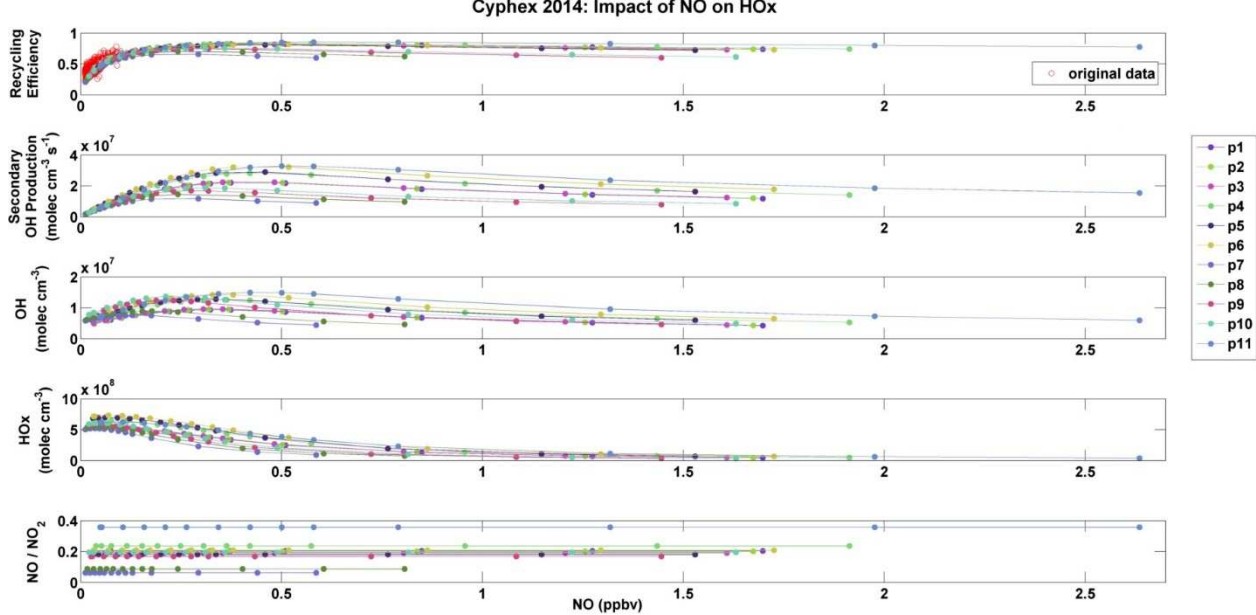

**Figure 8.** *Sensitivity of recycling efficiency, secondary OH production, OH and $HO_x$ to changes in $NO_x$ levels. P1-P11 are different starting points derived from the original dataset, for which $NO_x$ is ramped up to 3 ppbv, while maintaining the original NO-$NO_2$ ratio.*

## 6. Tables

**Table 1.** *Details of experiments conducted during CYPHEX (relevant to this study)* *Total uncertainty for the lower humidity range (< ca. 25% relative humidity)

| Species | Measurement Technique | Time Resolution | Precision | Limit of Detection | Total Uncertainty | Reference |
|---|---|---|---|---|---|---|
| **Hydroxyl Radical** | LIF-FAGE-IPI | 4 min | $4.8e5$ molec/cm$^3$ | $1e^6$ molec/cm$^3$ | 28.5% ($2\sigma$) | (Hens et al., 2014) |
| **Hydroperoxyl Radical** | LIF-FAGE | 14 sec | 0.4 pptv | 0.8 pptv | 36% ($2\sigma$) | (Martinez et al., 2010) |
| **Ozone** | UV Photometry | 1 min | | 2 ppbv | 5% | (Li et al., 2015) (Beygi, 2011) |
| **Carbon monoxide** | Room Temperature Quantum Caascade Laser spectrometer | 1 sec | | 0.4 ppbv | 14.4% | (Li et al., 2015) |
| **Nitric Oxide Nitrogen Dioxide** | Chemiluminescence | 5 sec | | 5 pptv 20 pptv | 20% 30% | (Li et al., 2015) (Beygi, 2011) |
| **Nitrous Acid** | Long Path Absorption Photometry | 30 sec | | 4 pptv | 10% | (Meusel et al., 2016) |
| **Formaldehyde** | Hantzsch reaction fluorescence | 10 min | | 38 pptv | 16% | (Kormann, 2003) |
| **Hydrogen peroxide Organic peroxides** | Enzymatic reaction | 10 min | | 14.9 pptv | 16% 20% | (Fischer, 2015) |
| **Methyl hydroperoxide** | High Performance Liquid Chromatography | 12 min | 3% | 25 pptv | 9% | Hafermann et al. (in prep) |
| **SO$_2$** | Chemical Ionization Mass Spectrometer | 10 mins | | 50 pptv | 30% | |
| **C$_2$-C$_4$ (alkanes, alkenes)** | GC-FID | 60 min | <5% | 1-8 pptv | 10% | (Sobanski et al., 2016) |
| **Methane** | GC-FID | 1 min | 2% | 20 ppbv | 2% | This Study |

| | | | | | | |
|---|---|---|---|---|---|---|
| **Isoprene** | GC-MS | 45 min | 3.3 % | 1ppt | 14.5% | (Derstroff et al., 2017) |
| **α-pinene** | | | 4.9 % | 1ppt | 15% | |
| **β-pinene** | | | 8.8% | 2ppt | 16.7% | |
| **Limonene** | | | 4.2% | 1ppt | 14.7% | |
| | PTR-TOF-MS | 1 min | 1 σ | 3σ | 1 σ | (Derstroff et al., 2017) |
| **Benzene** | | | 5.4% (at ~280 pptv) | 14 pptv | 14% (21%*) | |
| **Toluene** | | | 4.9% (at ~ 280 pptv) | 12 pptv | 14% (20%*) | |
| **Isoprene oxidation products** | | | 5.0% (at ~ 260 pptv) | 14 pptv | 11% (14%*) | |
| **MEK** | | | 3.8% (at ~ 280 pptv) | 16 pptv | 11% (16%*) | |
| **Methanol** | | | 2.5% (at ~2800pptv) | 242 pptv | 37% (41%*) | |
| **Acetaldehyde** | | | 2.2% (at ~ 1300 pptv) | 85 pptv | 22% (27%*) | |
| **Acetone** | | | 1.4% (at ~ 2500 pptv) | 97 pptv | 10% (17%*) | |
| **Acetic Acid** | | | 9.2 % (at ~ 900 pptv) | 264 pptv | 51% | |

**Table 2.** *Deposition velocities of some important species considered in the model scheme.*

| *Species* | *Deposition Velocity (cm s$^{-1}$)* |
|---|---|
| *Peroxides* | *4* |
| *PAN* | *0.2* |
| *NO$_3$* | *4* |
| *HNO$_3$* | *4* |
| *PINAL* | *0.6* |
| *HCOOH* | *1* |

### 7. Supplementary

Supplementary Figure 1: *Variation of atmospheric and background OH radicals during CYPHEX-2014.*

**Supplementary Figure 2a**. *Evolution of the OH signal in the low pressure detection cell with increasing residence time at 2 different NO concentrations (high NO: 1.71 x $10^{14}$ $cm^{-3}$ shown in blue; low NO: 7.1 x $10^{12}$ $cm^{-3}$ shown in red). The initial $HO_2$ and $RO_2$ signals are 10.7 and 18.1 pptv respectively.* **2b**. *Estimated $RO_2$ interference from all $RO_2$ and only from Isoprene based $RO_2$ during CYPHEX-2014 as function of the terpene concentrations.*

Supplementary Figure 3: *Source regions identified by FlexPart.*

**Supplementary Figure 4.** *LIF-FAGE measurements of OH (top panel) and $HO_2$ (bottom panel) vs model simulations with CAABA/MECCA. From left to right: model simulations using base case i.e. initialized with all measured species (case III), simulations emulating the autoxidation scheme (case V), after increasing the rate coefficient of RO2 – R'O2 reactions close to the gas kinetic limit (case VI).*

**Supplementary Figure 5.** *Time series of LIF-FAGE measurements of OH (top panel) and $HO_2$ radicals (bottom panel) along with various model simulations with CAABA/MECCA; model simulations using base case i.e. initialized with all measured species (case III), simulations emulating the autoxidation scheme (case V), after increasing the rate coefficient of RO2 – R'O2 reactions close to the gas kinetic limit (case VI). Time is in UTC. Local time in Cyprus during summer is UTC+3.*

**Supplementary Figure 6.** *Variation of the difference between modelled and measured $HO_2$ radicals w.r.t. the 'other term' in the $HO_2$ radical loss budget, most of which is constituted by the reactions of peroxy alkyl and acyl radicals with $HO_2$.*

Supplementary File1: Caaba/Mecca model scheme based on the Mainz Organic Mechanism (MOM).