# Peer review of "Oxidation processes in the Eastern Mediterranean atmosphere: Evidence from the Modelling of HOx Measurements over Cyprus"

_Atmospheric Chemistry and Physics, 2018_

## Referee Comment (RC1) · Anonymous Referee #1 · 12 Mar 2018

General comments:

This manuscript describes the analysis of a measurement campaign in Cyprus with an emphasis on understanding the OH and HO2 measurements there. The measurement suite seems to be fairly complete and the model is well documented. The analysis is fairly thorough, the conclusions are justified, and the citations are appropriate. The manuscript meets ACP standards. I recommend that it be published with (very) minor revisions.

Before the authors get too concerned about 17% differences between the measured and modeled HO2, they need to assess what the model uncertainty is. I suspect that

it is about (15-30)% at the $1\sigma$ confidence level. When that uncertainty is combined with the measurement uncertainty, why do the authors think that this 17% difference is significant?

That being said, I commend the authors for their approach to searching for a cause of this difference, whether it is meaningful or not. Examining the trend in deviation between the measured and modeled HO2 with terpene concentration is a good approach to looking for possible causes.

Specific comments: Abstract, Line 25. "The model simulations for OH showed very good agreement with in-situ OH observations. Model simulations for HO2 also agreed fairly well with in-situ observations except when pinene levels exceeded 80 pptv." Please be more quantitative, such as agree to within uncertainties of . . . Section 2.2, page 6, line 39. By heating the water source to 80C, I am surprised that you do not get condensation in the downstream lines, which are probably at 25-35C.

Section 3.2. I would like to know what motivated the authors to want to see if simple chemistry with no NMHCs would replicate their measurements, since they and others have often shown that it cannot in environments where NMHCs are present.

Section 3.4. What would happen if the authors greatly increased the reaction rates of RO2 – R'O2 reactions to close to gas kinetic? Would it have the same effect as lowering the RO2+HO2 rate coefficient or the auto-oxidation rate?

Figure 8. In the second panel, is the secondary production really a few times 10-12 molecules cm-3 s-1? Is there a typo?

Page 17, line 31. You define the recycling efficiency as "the ratio of OH produced from secondary sources via reactions of HO2 with NO and O3 (R6-R7) to the OH produced from primary and secondary sources." But then you talk about recycling efficiencies in percent. Please use either the ratio or define the percentages. How is the recycling efficiency related to the more familiar chain length?

[Figure]

Technical corrections:

Abstract, line 25, and other places. In situ is Latin and therefore should not be hyphenated and should in italics.

Introduction, lines 5-10. You identify OH by its name, followed by its chemical formula, but do not do this for CO, CO2, NOx, SO2, HNO3, and H2SO4. I think you should be consistent and name all chemical species when you first introduce them.

Introduction, line 18. "its" refers to the subject of the main clause, which is "lifetime". You meant "its" to refer to "OH". I suggest rewriting as "Because OH reacts rapidly with . . .., its lifetime . . ."

Introduction, line 11. "upto" should be "up to".

Section 1.2, line 14. "Photochemical" should now be one word with no hyphen.

Section 1.2, line 18. ". . . still substantial having significant . . ."? Chose one or the other.

Section 2.1, page 5, line 1 (and other places). Directions such as "northwest" are one word and not hyphenated.

Page 9, line 17. "day to day" should be hyphenated when used as an adjective.

Page 10, line 5. The authors are pretty careful to distinguish between measured OH reactivity and calculated OH reactivity, and should do so on this line as well.

Page 10, line 11. "Suburban" is not hyphenated.

Page 11, line 42. Should be ". . ., which are sinks . . .".

Page 16, line 22. Should be "While, on average, the . . ."

Page 17, line 25. Would be better to use words: ". . . NO mostly less than 100 pptv. . .".

---

## Referee Comment (RC2) · Anonymous Referee #2 · 28 Mar 2018

This paper presents measurements of OH and HO2 radical concentrations over Cyprus during the CYprus PHotochemistry EXperiment (CYPHEX) in 2014. Measurements of OH and HO2 using an LIF-FAGE instrument were compared to the results of a box model constrained by observations of O3, CO, NOx, hydrocarbons, and others. The model used agrees well with the measured OH to within 10% when constrained to all measured species, including NMHC, but tends to underestimate HO2 by 17%. Excluding terpenes from the model improves the modeled agreement with the measured HO2. Increasing the concentration of terpenes was found to increase the modeled underprediction of the measured HO2. The authors suggest that the reason for the discrepancy is an increase in radical-radical termination reactions from terpene-generated

organic peroxy radicals. Adding a simple autooxidation scheme that decreases the terpene peroxy radical concentration improves the modeled agreement with the measured HO2.

The paper presents some interesting results that are appropriate for ACP and will be suitable for publication after the authors have addressed the following questions.

1) The authors state that an OH artifact generated internally on average accounted for approximately 45% of the total signal (page 6). While a detailed analysis of the interference is beyond the scope of the paper, the authors should expand the description of the interference and its variability. Unfortunately, the figure in the supplement does not provide much information on the day-to-day variability of the interference given that the background signal is displayed in arbitrary units. Can the authors display the interference as an equivalent OH concentration together with the total measured signal and the atmospheric OH signal as they have done previously (Novelli et al., 2014a)? Did this interference vary with ozone and biogenic VOC concentrations, as observed previously?

2) To reduce the potential interference in their HO2 measurements from RO2 radicals, the authors reduce the conversion efficiency of HO2 to OH to 30% (page 6). It is not clear whether the authors calibrated the conversion efficiency of RO2 radicals such as isoprene-based hydroxy peroxy radicals at this HO2-to-OH conversion efficiency to insure that potential interferences from RO2 radicals were indeed minimized. This should be clarified in the revised manuscript.

3) While the paper describes the range of observed concentrations of some of the other chemical parameters, it would be useful to show the time series of the measurements to illustrate their day-to-day variability and allow a comparison with the OH and HO2 measurements. Given the dependence of the model on the concentration of terpenes, the authors should also show the time series of the ambient isoprene and terpene concentrations to allow a comparison with the OH and HO2 measurements. A time

series of the NO2 mixing ratios would also be useful to allow comparisons with other environments.

4) While the correlation plots provide an overall picture of the agreement of the model with the measurements, they do not provide any information regarding the day-to-day variability of the model-measurement agreement as well as the ability of the model to reproduce the diurnal variations of OH and HO2. The authors should illustrate the modeled time series of OH and HO2, perhaps illustrating periods of better model agreement when measured terpene concentrations were lower than 80 ppt.

5) The description of the MOM chemical mechanism used by the authors gives a reference of Taraborrelli et al., 2015 (page 7), which is not in the reference list. Have the authors updated the MIM3 mechanism described in Taraborrelli et al. (2012) reflecting the updated LIM1 mechanism (Peeters et al., J. Phys. Chem. A, 118, 8625−8643, 2014) and used in the latest version of the MCM (Jenkin et al., Atmos. Chem. Phys., 15, 11433–11459, 2015)? While recycling of HOx radicals by isoprene may not be important in this environment, the isoprene mechanism used in the model should be clarified.

6) While Figure 5 illustrates the improved agreement with measurements when a simplified autooxidation scheme in the terpene oxidation mechanism is added, there is little discussion of the resulting modeled OH concentrations. Does the increase in the modeled HO2 lead to an increase in the modeled OH? Did the authors make any assumptions regarding the fate of the products of the H-shift reactions (photolysis, etc.)?

7) The radical budget illustrated in Figure 7 are difficult to read, especially the contributions to OH loss. While O3 and HONO photolysis and recycling by HO2 + O3 and NO contribute to approximately 80% of OH production (page 16), what contributes to the remaining 20% ("other" in Figure 7)? Ozonolysis? OH recycling from isoprene? For the loss of HO2, the "other" category appears to be a significant contribution on several days - is this category due to RO2 +HO2 reactions from terpenes as discussed in section 3.3 and 3.4? Is the agreement with the measured HO2 better on the days when this "other" loss is small? The paper would benefit from some additional discussion of the radical budget.

---

## Author Comment (AC1) · 7 Jun 2018

General comments:

This manuscript describes the analysis of a measurement campaign in Cyprus with an emphasis on understanding the OH and HO2 measurements there. The measurement suite seems to be fairly complete and the model is well documented. The analysis is fairly thorough, the conclusions are justified, and the citations are appropriate. The manuscript meets ACP standards. I recommend that it be published with (very) minor revisions.

>>Thank you very much for your positive comments and encouragement. We have provided a point-to-point reply to your other comments below.

Before the authors get too concerned about 17% differences between the measured and modeled HO2, they need to assess what the model uncertainty is. I suspect that it is about (15-30)% at the 1σ confidence level. When that uncertainty is combined with the measurement uncertainty, why do the authors think that this 17% difference is significant? That being said, I commend the authors for their approach to searching for a cause of this difference, whether it is meaningful or not. Examining the trend in deviation between the measured and modeled HO2 with terpene concentration is a good approach to looking for possible causes.

Response:-We have now assessed the model uncertainty using Monte-Carlo simulations. The simulations are carried out by selecting 228 data points from the input data set representing at least two occurrences of concentrations i.e. 1-15, 25-75 and 85-99 percentiles of $O_3$, CO, NO, HCHO, HONO, $C_5H_8$, $C_{10}H_{16}$ and $J(O^1D)$. For each of these points, we performed 9999 Monte Carlo simulation runs. The Monte Carlo simulations are based on the method described in *Sander et. al.* (2011) but additionally we varied the boundary conditions of the measured species by their $\pm 1$ σ uncertainty (Table 1 in MS). The derived overall model uncertainty at 1 σ is 17.4 % and 10.5 % for OH and $HO_2$ respectively.

Using a York fit (York et al., 2004) accounting for uncertainties in both model and observations, we found that the slope of the best fit line for model predicted $HO_2$ normalized to observed $HO_2$ with increasing terpene levels for the base case in figure 6 is significantly different from zero (with a probability of greater than 99.9 %) . That was the reason we further probed into the modelled $HO_2$ loss with increasing terpene levels. We thank you again for your encouragement towards the same.

York, D., Evensen N., Martinez, M., Delgado, J.  (2004) Unified equations for the slope, intercept, and standard errors of the best straight line, *Am. J. Phys.* **72 (3)**.

Changes in MS:-

Page 13: To access the model uncertainty, Monte-Carlo simulations are carried out by selecting 228 data points from the input data set representing at least two occurrences of concentrations i.e. 1-15, 25-75 and 85-99 percentiles of $O_3$, CO, NO, HCHO, HONO, $C_5H_8$, $C_{10}H_{16}$ and $J(O^1D)$. For each of these points, we performed 9999 Monte Carlo simulation runs. The Monte Carlo simulations are based on the method described in *Sander et. al.* (2011) but additionally we varied the boundary conditions of the measured species by their $\pm 1$ σ uncertainty (Table 1). The derived overall model uncertainty at 1 σ is 17.4 % and 10.5 % for OH and $HO_2$ respectively.

Specific comments: Abstract, Line 25. "The model simulations for OH showed very good agreement with in-situ OH observations. Model simulations for HO2 also agreed fairly well with in-situ observations except when pinene levels exceeded 80 pptv." Please be more quantitative, such as agree to within uncertainties of . . .

Response:-We have now modified the abstract as mentioned below.

Changes in MS:-

Abstract: - The model simulations for OH agreed to within 10 % with *in situ* OH observations. Model simulations for $HO_2$ agreed to within 17 % of the *in situ* observations. However, the model strongly underpredicts $HO_2$ at high terpene concentrations, this underprediction reaching up to 38 % at the highest terpene levels.

Section 2.2, page 6, line 39. By heating the water source to 80C, I am surprised that you do not get condensation in the downstream lines, which are probably at 25-35C.

Response: - We are sorry, this was a mistake in the write-up, the temperature was below the dew point in the container where the humidification device was placed between 25-30˚C.

Changes in MS:-

Page 8: The humidified air was generated by bubbling dry air into a container half-filled with water maintained at 25-30˚C.

Section 3.2. I would like to know what motivated the authors to want to see if simple chemistry with no NMHCs would replicate their measurements, since they and others have often shown that it cannot in environments where NMHCs are present.

Response: - We have two reasons for this:

a)  As the OH reactivity measurements had not been significantly above their detection limit, we have to deal with uncertainty if all important NHMCs are measured. By performing a run without any NHMC this run should give a frames to the maximum uncertainty for OH and $HO_2$ when non measured NHMC are present but not accounted for.

b)  Global models try to keep a minimum of chemical reactions and species as computations become expensive. To demonstrate the contribution of different level of NHMC for HOx levels in this environment, we introduced the NHMC by increasing complexity and likelihood that they are considered in global models.

Section 3.4. What would happen if the authors greatly increased the reaction rates of RO2 – R'O2 reactions to close to gas kinetic? Would it have the same effect as lowering the RO2+HO2 rate coefficient or the auto-oxidation rate?

Response: - We thank you very much for this suggestion. Actually, when we greatly increased the $RO_2$-$R'O_2$ reaction rate to $4e^{-10}$ $cm^3$ $molecule^{-1}$ $s^{-1}$ (close to the collision limit), we found that the discrepancy of the modelled $HO_2$ loss w.r.t. the measurements with increasing terpenes decreases and the overall agreement between modelled and measured $HO_2$ is still good.

Changes in MS:-

Figure 6 modified to add result of simulation where the $K_{RO_2R'O_2}$ was increased close to collision limit.

[Figure]

**Figure 6.** *$HO_2$ predictions using different chemical schemes which were employed to investigate the impact of peroxy radicals as $HO_2$ sink. The x-axis shows the sum of measured α-pinene, β-pinene and limonene. The base case is the model simulation using measured species; $4*K_{RO_2NO}$ is a simulation where the rates of selected terpene-related $RO_2$ species with NO are increased by a factor of 4; $0.2*K_{RO_2HO_2}$ is a simulation where the rates of these $RO_2$ species with $HO_2$ are reduced by a factor of 0.2; the autoxidation simulation includes an autoxidation scheme rapidly converting selected $RO_2$ species into ROOH.* *The last scheme refers to a simulation where $K_{RO_2R'O_2}$ is set to $4e^{-10}$ cm$^3$ molecule$^{-1}$ s$^{-1}$ (close to the collision limit).* *The mean and sigma are calculated for 100 pptv bins along the x-axis.*

Page 17: Further, when we greatly increased the $RO_2$-$R'O_2$ reaction rates to $4e^{-10}$ cm$^3$ molecule$^{-1}$ s$^{-1}$ (close to the collision limit), we found that the discrepancy of the modelled $HO_2$ loss w.r.t. the measurements with increasing terpenes decreases (Figure 6) and the overall agreement between modelled and measured $HO_2$ is still good (Figure 6, Supplementary Figure 4 & 5).

[Figure]

**Supplementary Figure 4.** *LIF-FAGE measurements of OH (top panel) and HO₂ (bottom panel) vs model simulations with CAABA/MECCA. From left to right: model simulations using base case i.e. initialized with all measured species (case III), simulations emulating the autoxidation scheme (case V), after increasing the rate coefficient of RO2 – R'O2 reactions close to the gas kinetic limit (case VI).*

[Figure]

**Supplementary Figure 5.** *Time series of LIF-FAGE measurements of OH (top panel) and HO$_2$ (bottom panel) along with various model simulations with CAABA/MECCA; model simulations using base case i.e. initialized with all measured species (case III), simulations emulating the autoxidation scheme (case V), after increasing the rate coefficient of RO2 – R'O2 reactions close to the gas kinetic limit (case VI). Time is in UTC. Local time in Cyprus during summer is UTC+3.*

Figure 8. In the second panel, is the secondary production really a few times 10-12 molecules cm-3 s-1? Is there a typo?

Response:-Thanks for pointing it out, this was a mistake as I forgot to convert the mixing ratio to number density. This is rectified now.

Changes in MS:- Figure 8 is now modified as per comment (panel 2).

[Figure]

**Figure 8.** *Sensitivity of recycling efficiency, secondary OH production, OH and $HO_x$ to changes in $NO_x$ levels. P1-P11 are different starting points derived from the original dataset, for which $NO_x$ is ramped up to 3 ppbv, while maintaining the original $NO$-$NO_2$ ratio.*

Page 17, line 31. You define the recycling efficiency as "the ratio of OH produced from secondary sources via reactions of HO2 with NO and O3 (R6-R7) to the OH produced from primary and secondary sources." But then you talk about recycling efficiencies in percent. Please use either the ratio or define the percentages. How is the recycling efficiency related to the more familiar chain length?

Response:- Thanks for pointing this out. We have now replaced the percentages with ratios. Further, the formula for recycling efficiency is corrected by adding the contribution from the photolysis of organic peroxides and correcting for the primary OH production from HONO by subtracting off the recombination reaction of OH with NO. The chain length was calculated to be 2.8 for the maximum recycling efficiency of 0.7 observed for CYPHEX using the formula (recycling efficiency)$^n$=1/e; where n is the chain length, assuming the OH will recycle until it reaches 1/e of its initial value.

Changes in MS:- At all places (Abstract, pages 19-21) where recycling efficiency was mentioned in percentage, it is replaced by the ratio.

Page 19-20: Added text (red portions changed)

The recycling efficiency for OH (REf-OH) is defined as the ratio of OH produced from secondary sources via reactions of $HO_2$ with $NO$ and $O_3$ (R6-R7) to the OH produced from primary and secondary sources. The primary sources of OH considered in our calculation include reaction of $H_2O$ with $O^1D$ produced from $O_3$ photolysis and the photolysis of HONO, $H_2O_2$ and organic peroxides. The contribution of the primary OH production from HONO photolysis is corrected by subtracting off the contribution from the recombination reaction of OH with NO.

While the present study is limited to NO <100 pptv, modelling results show that REf-OH increases with increase in NO levels (Figure 8). The REf-OH increases from 0.28 at 10 pptv to 0.42 at around 30 pptv NO

to 0.7 at 100 pptv NO (Figure 8). The REf-OH of 0.7 corresponds to a chain length of 2.8. The relationship of REf-OH with NO indicates that at higher NO (not observed for the CYPHEX data), the REf-OH would increase further. To understand if REf-OH would reach runaway conditions or saturate at a particular value, we chose several points along the REf-OH-NO curve and for each of these points, we generated hypothetical input data set keeping all parameters constant but changing the $NO_x$ levels from 0.2 to 3 ppbv, while maintaining the original $NO/NO_2$ ratio (Figure 8).

Figure 8 shows that in general the REf-OH increases to about 0.85 at 500 pptv NO. After this, it decreases gradually, related to the reaction of $NO_2$ with OH. Concurrently, the loss of $HO_2$ due to self-reactions and reaction of $HO_2$ with $CH_3O_2$ becomes much weaker compared to the loss of $HO_2$ to NO with increasing NO levels. The ratio of $HO_2$ losses via the radical-radical interactions (i.e. the reactions of $HO_2$ with $HO_2$ and $CH_3O_2$) to the $HO_2$ losses via the recycling channel (i.e. reactions of $HO_2$ with NO and $O_3$) decreases exponentially becoming less than 1 at 35 pptv NO and less than 0.5 at 70 pptv NO and 0.01 at 1 ppbv NO. The secondary OH production also shows a similar pattern with respect to REf-OH, peaking around 500 pptv for the different input data sets and decaying thereafter. The model generated OH peaks in this range but the $HO_x$ concentrations start to drop much earlier, from around 70 pptv NO. At these low NO levels, $HO_x$ loss by peroxide-peroxide reactions is comparable to the channels recycling $HO_2$ into OH, which can then be removed through reaction e.g. with $NO_2$. However, the recycling efficiency is only about 0.6 at 70 pptv NO. Budget analysis indicates that $HO_x$ production drops with increasing NO levels for this test dataset constrained to specific $NO_x$ levels. Overall, for the hydrocarbon levels observed during CYPHEX, the self-cleaning capacity of the atmosphere peaks at around 500 pptv NO, which at a $NO/NO_2$ ratio of 0.2 is equivalent to about 3.0 ppbv $NO_x$. Increasing $NO_x$ emissions further would lead to decreased $HO_x$ levels and the recycling efficiency is unlikely to increase further. On the other hand, decreasing NO levels below a few hundred pptv of NO would also decrease the recycling efficiency for the same hydrocarbon levels indicating that small amount of $NO_x$ help sustain the self-cleaning capacity of the atmosphere (Figure 8).

Technical corrections:

Abstract, line 25, and other places. In situ is Latin and therefore should not be hyphenated and should in italics.

Response:- Done as suggested.

Introduction, lines 5-10. You identify OH by its name, followed by its chemical formula, but do not do this for CO, CO2, NOx, SO2, HNO3, and H2SO4. I think you should be consistent and name all chemical species when you first introduce them.

> Response:- Done as suggested.

Introduction, line 18. "its" refers to the subject of the main clause, which is "lifetime". You meant "its" to refer to "OH". I suggest rewriting as "Because OH reacts rapidly with . . .., its lifetime . . ."

Response:- Done as suggested.

Introduction, line 11. "upto" should be "up to".

Response:- Done as suggested.

Section 1.2, line 14. "Photochemical" should now be one word with no hyphen.

Response:- Done as suggested.

Section 1.2, line 18. ". . . still substantial having significant . . ."? Chose one or the other.

Response:- Done as suggested.

Section 2.1, page 5, line 1 (and other places). Directions such as "northwest" are one word and not hyphenated.

Response:- Done as suggested.

Page 9, line 17. "day to day" should be hyphenated when used as an adjective.

Response:- Done as suggested.

Page 10, line 5. The authors are pretty careful to distinguish between measured OH reactivity and calculated OH reactivity, and should do so on this line as well.

Response:- Done as suggested.

Page 10, line 11. "Suburban" is not hyphenated.

Response:- Done as suggested.

Page 11, line 42. Should be ". . ., which are sinks . . .".

Response:- Done as suggested.

Page 16, line 22. Should be "While, on average, the . . ."

Response:- Done as suggested.

Page 17, line 25. Would be better to use words: ". . . NO mostly less than 100 pptv. . .".

Response:- Done as suggested
* * *
**Anonymous Referee #2**

This paper presents measurements of OH and HO2 radical concentrations over Cyprus during the CYprus PHotochemistry EXperiment (CYPHEX) in 2014. Measurements of OH and HO2 using an LIF-FAGE instrument were compared to the results of a box model constrained by observations of O3, CO, NOx, hydrocarbons, and others. The model used agrees well with the measured OH to within 10% when constrained to all measured species, including NMHC, but tends to underestimate HO2 by 17%. Excluding terpenes from the model improves the modeled agreement with the measured HO2. Increasing the concentration of terpenes was found to increase the modeled underprediction of the measured HO2. The authors suggest that the reason for the discrepancy is an increase in radical-radical termination reactions from terpene-generated organic peroxy radicals. Adding a simple autooxidation scheme that decreases the terpene peroxy radical concentration improves the modeled agreement with the measured HO2.

The paper presents some interesting results that are appropriate for ACP and will be suitable for publication after the authors have addressed the following questions.

>> Thank you very much for your positive comments and encouragement. We have provided a point-to-point reply to your other comments below.

1) The authors state that an OH artifact generated internally on average accounted for approximately 45% of the total signal (page 6). While a detailed analysis of the interference is beyond the scope of the paper, the authors should expand the description of the interference and its variability. Unfortunately, the figure in the supplement does not provide much information on the day-to-day variability of the interference given that the background signal is displayed in arbitrary units. Can the authors display the interference as an equivalent OH concentration together with the total measured signal and the atmospheric OH signal as they have done previously (Novelli et al., 2014a)? Did this interference vary with ozone and biogenic VOC concentrations, as observed previously?

Response:- We have now modified the plot as per comments. We have, however, not shown the total OH signal, as it can be estimated from the sum of background and atmospheric signals, hence redundant. We do not see a clear variation of the background OH with either $O_3$ or terpenes alone. However, when we plot the background OH with a proxy for Crigee (CI) production rate, there seems to be some relationship at low and moderate CI concentrations, which needs to be studied in detail. This will be attempted in a future work as CI is not within the scope of the present MS as the reviewers themselves point out.

[Figure]

Figure. Background OH plotted against a simplified proxy for crigee (CI) concentrations, assuming that the major production term for CI is the ozonolysis of terpenes and the major loss term is its reaction with water dimer. While the $O_3$ and terpene concentrations are based on measurements, the concentration of $H_2O$ dimer is based on the Caaba/Mecca output.

Changes in MS:- Supplementary figure 1 modified

[Figure]

**Supplementary Figure 1**: *Variation of atmospheric and background OH during CYPHEX.*

2) To reduce the potential interference in their HO2 measurements from RO2 radicals, the authors reduce the conversion efficiency of HO2 to OH to 30% (page 6). It is not clear whether the authors calibrated the conversion efficiency of RO2 radicals such as isoprene-based hydroxy peroxy radicals at this HO2-to-OH conversion efficiency to insure that potential interferences from RO2 radicals were indeed minimized. This should be clarified in the revised manuscript.

Response:- Due to low OH reactivity during Cyphex, we expect the $RO_2$ production from the oxidation of hydrocarbons to be low. Hence the interference due to potential conversion of atmospheric $RO_2$s into $HO_2$ due to NO injected to convert $HO_2$ into OH in the low pressure detection volume is expected to be low compared to regions with high OH reactivities like the boreal forest in Finland. In order to reduce the conversion of $RO_2$ to $HO_2$, we used a reduced NO flow resulting in ~7 x $10^{12}$ $cm^{-3}$ of NO in the detection cell, thus converting only about 30% of $HO_2$ to OH while simultaneously reducing the $RO_2$-$HO_2$ conversion efficiency. We did not conduct experiments to measure the conversion efficiencies of all possible $RO_2$s in the atmosphere. To estimate the interference due to $RO_2$s on our measured signal, we made model calculations using CAABA/MECCA where most of the $RO_2$s from higher hydrocarbons directly form $HO_2$ on reacting with NO skipping the reaction step of alkoxy radicals with $O_2$ which is slower at reduced pressure inside the instrument compared to ambient.

The model is run at ~4 hPa to see how OH and $HO_2$ evolve with time in the low pressure detection volume at different NO concentrations and is validated for calibration conditions (manuscript under preparation). The $RO_2$s in the model are initialized with the concentrations generated from the model run for the base case (case III in Figure 3 of the original MS) for our study while OH, $HO_2$ are initialized with measured concentrations.

[Figure]

**Supplementary Figure 2a**: Internal evolution of the OH signal in the low pressure detection cell with increasing residence time at 2 different NO concentrations (high NO: $1.71 \times 10^{14}$ cm$^{-3}$ shown in blue; low NO: $7.1 \times 10^{12}$ cm$^{-3}$ shown in red). The initial HO$_2$ and RO$_2$ signals are 10.7 and 18.1 pptv respectively.

In the **Supplementary Figure 2a,** we show the evolution of OH and HO$_2$ inside the detection cell after injection of NO (t=0) for NO concentrations of $7.1 \times 10^{12}$ and $1.71 \times 10^{14}$ cm$^{-3}$ respectively. The converted OH signal is detected after 6.6 msecs (time of detection). For the high conversion efficiency case, the contribution of the RO$_2$ to the OH signal at 6.6 ms is about 35% or 31% of the initial RO$_2$ mixing ratio. This value matches with estimates from previous study by Hens et al., 2014. For the low conversion efficiency case that represents the CYPHEX measurement mode, the estimated contribution of RO$_2$ to the measured signal is about 12 % or 2.5% of the initial RO$_2$ mixing ratio.

[Figure]

**Supplementary Figure 2b**: Estimated $RO_2$ interference from all $RO_2$ and only from Isoprene based $RO_2$ during CYPHEX-2014 as function of the terpene concentrations

Further, we estimate that more than 50% of the $RO_2$ interference is due to isoprene oxidation products. This is due to the fast conversion of isoprene-based hydroxy peroxy radicals towards $HO_2$ and OH. These isoprene-based hydroxy peroxy radicals have one of the largest conversion efficiencies of up to 90 % (Fuchs et al., 2011, Lew et al., 2018). Moreover, we see that the $RO_2$ interference does not increase with increasing terpene concentrations and is nearly constant at terpene levels greater than 80 pptv (**Supplementary Figure 2b**), mostly because during the course of the day $HO_2$ concentration increases during this time faster than terpene based $RO_2$ concentration. This indicates that the $RO_2$ interference effects cannot explain the deviation of modelled $HO_2$ w.r.t. measurements at high terpene mixing ratios.

Lew, M. M., Dusanter, S., and Stevens, P. S.: Measurement of interferences associated with the detection of the hydroperoxy radical in the atmosphere using laser-induced fluorescence, *Atmospheric Measurement Techniques*, **11** (1), 95-109, Doi 10.5194/amt-11-95-2018, 2018.

Changes in MS:- Added the above text in Page 8 and the figures as supplementary

3) While the paper describes the range of observed concentrations of some of the other chemical parameters, it would be useful to show the time series of the measurements to illustrate their day-to-day variability and allow a comparison with the OH and HO2 measurements. Given the dependence of the model on the concentration of terpenes, the authors should also show the time series of the ambient isoprene and terpene concentrations to allow a comparison with the OH and HO2 measurements. A time series of the NO2 mixing ratios would also be useful to allow comparisons with other environments.

Response:-We have now modified figure 2 as per suggestions.

Changes in MS:- Modified figure 2 as per suggestion

[Figure]

**Figure 2.** *Measurements of OH, HO₂ along with selected chemical and radiation parameters relevant to HO_x chemistry during the CYPHEX campaign. Time is in UTC. Local time in Cyprus during summer is UTC+3.*

4) While the correlation plots provide an overall picture of the agreement of the model with the measurements, they do not provide any information regarding the day-to-day variability of the model-measurement agreement as well as the ability of the model to reproduce the diurnal variations of OH and HO2. The authors should illustrate the modeled time series of OH and HO2, perhaps illustrating periods of better model agreement when measured terpene concentrations were lower than 80 ppt.

Response:-We have added the following figure as suggested.

Changes in MS:- Added text on Page 17 and supplementary figure 5 as per suggestion

The large deviation between observations and modelled HO₂ around middays on 205, 208 and 210 occur during periods with high terpene concentrations. On day 205 and 208, the simulation with the autoxidation scheme shows much better agreement to observations compared to the base case (Case III in figure 3) while on day 210, when terpene concentrations are much higher, even the autoxidation scheme fails to reproduce the HO₂ observations. In this case the simulation where the rate coefficient of RO₂-R'O₂ reactions was drastically increased shows a much better agreement compared to both the base case and the autoxidation case.

[Figure]

**Supplementary Figure 5.** *Time series of LIF-FAGE measurements of OH (top panel) and HO₂ (bottom panel) along with various model simulations with CAABA/MECCA; model simulations using base case i.e. initialized with all measured species (case III), simulations emulating the autoxidation scheme (case V), after increasing the rate coefficient of RO2 – R'O2 reactions close to the gas kinetic limit (case VI). Time is in UTC. Local time in Cyprus during summer is UTC+3.*

5) The description of the MOM chemical mechanism used by the authors gives a reference of Taraborrelli et al., 2015 (page 7), which is not in the reference list. Have the authors updated the MIM3 mechanism described in Taraborrelli et al. (2012) reflecting the updated LIM1 mechanism (Peeters et al., J. Phys. Chem. A, 118, 8625–8643, 2014) and used in the latest version of the MCM (Jenkin et al., Atmos. Chem. Phys., 15, 11433–11459, 2015)? While recycling of HOx radicals by isoprene may not be important in this environment, the isoprene mechanism used in the model should be clarified.

Response:- The reference to Taraborrelli et al is an EGU conference abstract is towards the end of the reference list (page 25 of original submission). As per comments, we have now expanded the model description as described below.

Changes in MS:-Page 10

MOM has been first presented and used in the study by Lelieveld et al. (2016). It represents the gas-phase oxidation of more than 40 primarily emitted VOCs. The structure and the construction methodology mirrors the one of the MCM. The oxidation mechanism for aromatics has been presented in Cabrera-Perez et al., (2016). The terpene oxidation scheme includes previous developments for modelling HOx field measurements (Taraborrelli et al. 2012; Hens et al. 2014; Nölscher et al. 2014). Most of the known and/or proposed HOx-recycling mechanisms under low-NO conditions are taken into account.  Finally, isoprene chemistry follows to a large extent Peeters et al. (2014) and Jenkin et al. (2015) with modifications by Nölscher et al. (2014). Chemistry of the pinenes (monoterpenes) is a reduction of the MCM with modifications proposed in the past by Vereecken et al., (2007); Nguyen et al., (2009); Vereecken and Peeters, (2012); Capouet et al., (2008).

Cabrera-Perez, D., Taraborrelli, D., Sander, R., and Pozzer, A.: Global atmospheric budget of simple monocyclic aromatic compounds, Atmos. Chem. Phys., 16, 55 6931–6947, https://doi.org/10.5194/acp-16-6931-2016, http://www.atmos-chem-phys.net/16/6931, 2016.

Capouet, M., Müller, J.-F., Ceulemans, K., Compernolle, S., Vereecken, L., and Peeters, J.: Modeling aerosol formation in alpha-pinene photo-oxidation experiments, J. Geophys. Res., 113D, https://doi.org/10.1029/2007JD008995, 2008.

Jenkin, M. E., Young, J. C., and Rickard, A. R.: The MCM v3.3.1 degradation scheme for isoprene, Atmos. Chem. Phys., 15, 11 433–11 459, https://doi.org/10.5194/acp-15-11433-2015, 2015.

Nguyen, T. L., Peeters, J., and Vereecken, L.: Theoretical study of the gas-phase ozonolysis of b-pinene (C10H16), Phys. Chem. Chem. Phys., 11, 5643–5656, https://doi.org/10.1039/b822984h, 2009.

Nölscher, A., Butler, T., Auld, J., Veres, P., Muñoz, A., Taraborrelli, D., Vereecken, L., Lelieveld, J., and Williams, J.: Using total OH reactivity to assess isoprene photooxidation via measurement and model, Atmos. Environ., 89, 453–463, https://doi.org/10.1016/j.atmosenv.2014.02.024, 2014.

Peeters, J., Müller, J.-F., Stavrakou, T., and Nguyen, V. S.: 30 Hydroxyl radical recycling in isoprene oxidation driven by hydrogen bonding and hydrogen tunneling: the upgraded LIM1 mechanism, J. Phys. Chem. A,

Vereecken, L. and Peeters, J.: A theoretical study of the OH-initiated gas-phase oxidation mechanism of b-pinene (C10H16): first generation products, Phys. Chem. Chem. Phys., 14, 3802–25 3815, https://doi.org/10.1039/c2cp23711c, 2012.

Vereecken, L., Müller, J.-F., and Peeters, J.: Low-volatility poly-oxygenates in the OH-initiated atmospheric oxidation of a-pinene: impact of non-traditional peroxyl radical chemistry, Phys. Chem. Chem. Phys., 9, 5241–5248, 30, https://doi.org/10.1039/b708023a, 2007.

6) While Figure 5 illustrates the improved agreement with measurements when a simplified autooxidation scheme in the terpene oxidation mechanism is added, there is little discussion of the resulting modeled OH concentrations. Does the increase in the modeled HO2 lead to an increase in the modeled OH? Did the authors make any assumptions regarding the fate of the products of the H-shift reactions (photolysis, etc.)?

Response:-The autoxidation scheme used by us is based on Crounse et al (2013). In this scheme (Figure 1 in Crounse et al., 2013), the hydrogen shift reactions lead to generation of a hydroperoxide compound along with an OH. This directly leads to an increase in OH as can be observed from the supplementary figures 4 & 5. Since there is not sufficient information regarding the compounds that would be formed during the autoxidation of peroxy radicals that we considered for this analysis, we have used dummy species for the hydroperoxide products formed. These dummy species will not further participate in the chemical scheme.

Changes in MS:- Supplementary figures 4 & 5 added to show how both modelled OH and HO$_2$ vary against measurements for the autoxidation scheme w.r.t. the base case.

7) The radical budget illustrated in Figure 7 are difficult to read, especially the contributions to OH loss. While O3 and HONO photolysis and recycling by HO2 + O3 and NO contribute to approximately 80% of

OH production (page 16), what contributes to the remaining 20% ("other" in Figure 7)? Ozonolysis? OH recycling from isoprene? For the loss of HO2, the "other" category appears to be a significant contribution on several days - is this category due to RO2 +HO2 reactions from terpenes as discussed in section 3.3 and 3.4? Is the agreement with the measured HO2 better on the days when this "other" loss is small? The paper would benefit from some additional discussion of the radical budget.

Response:- We have now modified figure 7a so that the panel for the OH sink as well as the text look bigger and clearer. There was a slight mistake in the budget calculation whereby the reactions of the category ROOH + OH = product + OH; e.g.  'C96OOH + OH = NORPINAL + OH'; which are neither a OH source or sink, were not omitted. Now they taken out of the budget and the numbers in the text are revised, although the resulting changes are very minor and always less than 2 %. Further, more reactions are added to the legend in the plot of OH production and loss budget to reduce the 'others' part and these described in the text. The others category is now about only about 7 % compared to nearly 18 % previously. This 7% is made up of numerous reactions which contribute less than 1% individually, so not worthy of discussion here. Similarly, in the OH loss category we have added a group of reactions of the category ROOH + OH = RO2, e.g. 'C85OOH + OH = C85O2', which makes up about 8.4 %. The associated changes are documented in the text. Further, we have now shown the $HO_2$ sinks in much more detail by modifying figure 7b. The peroxy radicals together account for nearly 95% of the other term and described in the revised budget plot. You are right that the model increasingly underpredicts the measurements when 'other' loss is large and these occur at higher terpene concentrations (supplementary figure 6).

[Figure]

**Supplementary Figure 6.** *Variation of the difference between modelled and measured $HO_2$ w.r.t. the 'other term' in the last version of the MS, most of which is constituted by the reactions of peroxy alkyl and acyl radicals with $HO_2$.*

Changes in MS:- Text (**only the red portions**) and modified fig 7a

For CYPHEX, the major OH as well as $HO_x$ producing channel is the reaction of atmospheric water vapor with $O^1D$ generated from the photolysis of $O_3$ (Figure 7). Peak daytime contributions of this channel towards OH production exceeded 45 % for most of the days, and about 60 % on days 205, 208 and 209. The midday values coinciding with peak OH production on day 205 was marked by conspicuous influence of aged air masses originating over south-west Europe and considerably processed over the Mediterranean before reaching the site. The peak $HO_2$ values on this day were about 11% lower than the average peak $HO_2$ values during the study period. The peak $HO_2$ values for $J(O^1D)>2.5e^{-5}$ $s^{-1}$ was 6.4 x $10^8$ molec/$cm^3$ for the study period while this value was only 5.7 x $10^8$ molec/$cm^3$ on day 205. While, on average, the recycled OH via reactions of $HO_2$ with NO and $O_3$ (R6-R7) contributed 33.6 % to the total OH production, this value was about 6 % lower for days 205-206. It may also be noted that although $O_3$ was very low on both days 205 and 206, with predominant influence of aged marine air, the contribution from $O^1D+H_2O$ to the total OH production still exceeded 50 %. Due to lower HONO mixing ratios, the fractional contributions of HONO photolysis towards peak OH production during midday on day 208 was significantly low at about 2.5-3.5 %. On all other days, for which values are available, this channel contributed more than 6 % to peak OH production during noon time. The photolysis of HONO has the largest fractional contribution to the early morning OH production on day 211, reaching above 30 %. Overall during CYPHEX, the average daytime ($JO^1D>0$) contribution to OH production from $O_3$ photolysis and subsequent reaction of $O^1D$ with water vapor was about 39.1 %, the average daytime contribution from HONO photolysis was 12.3 %, while recycled OH from reaction of $HO_2$ with $O_3$ and NO account for 15.2 and 18.4 % of the total OH production, respectively. The four major OH producing channels (Figure 7) contribute up to 95 % of daytime OH production on most occasions, with 85.3 % on average. Further, the reactions of acyl peroxy radicals ($RCO_3$) with $HO_2$ contribute about 3.1 % to the OH production which the photolysis of $H_2O_2$ and the ozonolysis of pinene contribute 1.85 % and 2 % respectively.

The single major sink of OH during CYPHEX was CO, followed by $CH_4$, HCHO, $C_5H_8$, $CH_3CHO$ and $O_3$, on average accounting 20.9, 10.0, 7.8, 5.1, 4.9 and 4.1 % of OH losses respectively. The reactions of various peroxides with OH to form peroxy radicals e.g.R24 contribute 8.4 % to the OH loss. Further, oxidation of $CH_3OH$, pinal, C2-C4 alkenes, α-pinene, $HO_2$, $NO_2$, $CH_3O_2$ by OH contributed 2.7, 2.4, 2.4, 2.2, 1.6, 1.4, 1.4 % respectively. During these days (205-206), the modelled OH loss (Figure 7) as well as the calculated OH reactivity (Figure 4) were lowest of the study period. It is likely that the air masses arriving to the site were already much processed, spending considerable time over the Atlantic and Mediterranean, leading to depleted OH reactivity. During this period of marine influence, the contribution of long-lived gases to the daytime OH loss increased by about 15 % while the contribution of shorter lived gases like HCHO and $CH_3CHO$ decreased by 34 % and 39 % respectively.

Added text in page 19

The reactions of peroxy radicals ($RO_2$ and $RCO_3$) with $HO_2$ contribute 24.6% to the $HO_2$ loss, resulting in increased underprediction of $HO_2$ by the model with increasing terpene concentrations (Supplementary Figure 6).

|
[Figure]
 | + OH | → | | (R24, G48207 in Model) |
|---|---|---|---|---|
| C₈H₁₃O₄ (C86OOH) | | | C₈H₁₃O₄ (C86O2) | |

[Figure]

[Figure]

Figure 7 a & c.